# Gaps and drivers of global marine animal biodiversity from the surface to abyss

Hanieh Saeedi [1,2] ✉

With advances in global biodiversity data sharing, particularly following the Census of Marine Life, understanding of marine biodiversity has improved but remains incomplete. The Ocean Biodiversity Information System and Global Biodiversity Information Facility host over 150 million marine occurrence records, enabling reassessment of global biodiversity and data gaps. Here, we compile a quality-controlled dataset of ca. 48 million records covering 184,141 marine animal species, representing ~87% of accepted World Register of Marine Species and 91% of Ocean Biodiversity Information System taxa. Generalised Linear and Additive Models assess how geoecological drivers and human impact influence species richness while accounting for sampling effort and spatial autocorrelation across depth and taxa. Approximately 50% of the global ocean remains insufficiently sampled, with more than 160 million km² below 200 m lacking data. Sampling is biased toward developed regions, especially the North Atlantic, with major gaps in equatorial and Global South regions. Central tropical areas ($-5°$ to $5°$) contribute only <2.5% of global records, helping explain non-significant bimodal latitudinal patterns. Shallow-water richness is mainly associated with temperature, while deep-sea patterns relate to human impact (sampling intensity) and nitrate-driven remineralisation. These results highlight major global data gaps and the need for depth-explicit, bias-aware biodiversity assessment and monitoring to support conservation and the UN Ocean Decade.

The current marine biodiversity pattern reflects not only a showcase of the complex ecological and evolutionary processes that have shaped the ocean life over millions of years[1–3] but also the limitations imposed by sampling and taxonomy biases[4,5]. The history of marine biodiversity is a narrative of ecological events, adaptation, interaction, and dispersal from ancient microbial life to the recent diverse marine ecosystems[6–9]. Fossil-based studies suggest that a tropical peak and poleward decline in species richness have not been a persistent pattern throughout the Phanerozoic, but are restricted to cold (tropical peak) and warm (temperate peaks) intervals of the Palaeozoic and the last 30 million years[10,11]. During the Phanerozoic, three distinct phases of increasing biodiversity were noted by the emergence of a unique

Evolutionary Fauna (EF)[9]. Within these EFs, stable periods known as Ecological Evolutionary Units (EEUs) existed, but they were often interrupted by extinction events triggered by glaciations and extraterrestrial impacts[9,12]. These extinctions left many ecological niches vacant, enabling surviving species to adapt and evolve. Over approximately five million years, this process led to a remarkable ecological recovery, shaping the current patterns of marine biodiversity we see today[9,12]. However, our limited knowledge of marine life and their distribution patterns often misleads our interpretation of true biodiversity gradients, hotspots, and coldspots.

The world's oceans cover more than 70% of the Earth's surface and are the largest biome[13]. The oceans produce approximately half of

[1]Senckenberg Research Institute and Natural History Museum, Senckenberg Data and Modelling Centre and Department of Marine Zoology, Geobiodiversity Informatics, Senckenberganlage 25, 60325 Frankfurt am Main, Germany. [2]Goethe University Frankfurt. Department 15 - Life Sciences, Institute for Ecology, Evolution and Diversity, 60438 Frankfurt am Main, Germany. ✉e-mail: hanieh.saeedi@senckenberg.de

global primary production, with phytoplankton photosynthesis producing much of Earth's oxygen[13,14]. Although they are very vast, marine ecosystems are not homogeneous. Rather, they exhibit distinct biodiversity patterns and species distribution influenced by various physico-chemical and geological factors, sampling and taxonomy biases, and gaps in data and knowledge[15-17]. Despite 250 years of taxonomic studies, over 2.2 million marine species exist, yet around 90% remain undescribed[18]. Alarmingly, less than 3% of oceans are fully/highly monitored and protected[19], leaving most species at risk of extinction before they are discovered. Furthermore, many described species remain undigitised or are not shared in open-access databases, particularly in underrepresented regions such as the tropical latitudes[4]. However, significant advances in data sharing over the past decades have equipped us with invaluable data, enabling comprehensive knowledge gap analyses and identifying the main drivers of biodiversity patterns.

The Census of Marine Life (CoML) was launched in 2000 as a global scientific initiative aimed at assessing and understanding marine biodiversity globally[20]. CoML played a key role in the emergence of the Ocean Biodiversity Information System (OBIS) in response to the urgent need to compile, standardise, and share marine biodiversity data[5,21,22]. The OBIS is essential for addressing complex ocean challenges, such as climate change, pollution, habitat loss, and overfishing, which require systematic and informed solutions. Over 2,000 scientific publications cited OBIS over the last two decades, highlighting its importance in scientific communities[23]. OBIS also provides data to key science-policy platforms, such as the Intergovernmental Science-Policy Platform on Biodiversity and Ecosystem Services (IPBES), the Convention on Biological Diversity (CBD), the Conference of the Parties (COP), and the UN Ocean Decade, to facilitate informed decision-making. The tailored analyses of the shared biodiversity data thus enhance biodiversity monitoring and conservation by identifying knowledge gaps, uncovering hidden patterns, identifying key drivers, and providing insights into marine ecosystem complexities to support evidence-based conservation policies[24-27].

The exploration of latitudinal diversity gradients (LDGs) in marine and terrestrial ecosystems has long aimed to uncover biodiversity patterns and their drivers. For decades, the paradigm suggested a single peak of marine species richness at the equator[28-30]. However, a study reported a global bimodal pattern in marine razor clams[31], with peaks at 10° and 25° latitude rather than at the equator, likely driven by temperature[31]. The authors later analysed data from 27 studies and OBIS global datasets, revealing a similar bimodal pattern in all marine species, with peaks at mid-latitudes, particularly in the northern hemisphere[32]. They proposed that tropical species evolve near the tropics' edges, responding to temperature fluctuations and productive habitats. Research literature has suggested that the bimodal pattern in pelagic taxa likely emerged before the Anthropocene, ca. 15,000 y ago[11], largely controlled by temperature and metabolic tradeoffs[11,33,34]. In contrast, some argue that the dip in species richness near the equator may reflect insufficient sampling rather than true biodiversity loss[35-37]. They concluded that inadequate sampling can bias global species richness estimates, obscuring true latitudinal biodiversity patterns and their drivers.

This work integrates a comprehensive dataset while systematically dividing the global ocean into distinct depth zones (Supplementary Figs. 1–3) and taxonomic groups. Sampling biases were addressed using rarefaction-based species curves and standardised species richness estimates (ES50), which control for differences in sampling effort by estimating species richness at a fixed sample size of 50 occurrences. Given the substantial growth of digital marine data since our study in 2026[32], which documented 65,000 marine species in OBIS, the current total exceeds 180,000 accepted species in OBIS and GBIF, with over 47 million quality-controlled occurrence records. This expansion calls for a reassessment of global latitudinal species

richness patterns and their ecological and anthropogenic drivers. To model species richness responses to those drivers, a comprehensive set of ecological and bathymetric variables known to influence the physiology, biology, and distribution of marine species[38] was compiled from the Bio-ORACLE database. These variables, including physical (e.g., temperature, current velocity, bathymetry), chemical (e.g., oxygen, nitrate), and productivity-related parameters (e.g., primary productivity, radiation), were used across pelagic and benthic layers to capture bathymetrical environmental variability.

Multiple environmental drivers, such as temperature, dissolved oxygen, and primary productivity, are key determinants of species richness in shallow marine ecosystems[17,39,40]. In contrast, the deep sea is characterised by extreme conditions such as high pressure, low temperatures, and limited light availability, which impose strong constraints on deep-sea life[15,41,42]. Additionally, spatial features such as continental shelves and margins were derived from the Marine Regions datasets[43]. Spatial features are essential for marine biodiversity because they provide environmental conditions, habitats, and connectivity, resulting in shaping species distribution and richness. Anthropogenic pressure was also incorporated using the Global Human Influence Index (obtained from NASA as a 1-km resolution global dataset integrating population pressure, land use, infrastructure, and accessibility to quantify human impact)[44], enabling an integrated assessment of environmental and human drivers of biodiversity patterns. This approach reveals previously unrecognised patterns of marine biodiversity and knowledge gaps, providing critical insights for biodiversity monitoring, conservation planning, and sustainable ocean management in the face of accelerating environmental change.

In this work, a comprehensive, quality-controlled global dataset comprising approximately 48 million marine animal occurrence records covering 184,141 species (ca. 87% of accepted WoRMS species) across shallow, mesopelagic, and deep-sea habitats is presented (Supplementary Figs. 2 and 3). Global marine biodiversity patterns are mapped, and major data gaps are quantified, showing that around half of the ocean, particularly below 200 m depth, remains undersampled or lacks sufficient occurrence data. Occurrence records are strongly concentrated in developed regions such as the North Atlantic, while large gaps persist in the East Pacific and equatorial regions. For example, central tropical regions (−5° to 5°) remain disproportionately underrepresented despite improving data sharing, contributing less than 2.5% of OBIS records over the past decade. Reported species richness likely underestimates true biodiversity, especially in deep-sea, equatorial and polar regions with high potential for species discoveries. Consistent species richness hotspots are identified across depth zones, including the Gulf of Mexico, New Caledonia, and northern New Zealand, while bias-corrected analyses reveal additional, previously overlooked hotspots worldwide. Finally, distinct environmental drivers of species richness are shown across depth zones, with shallow-water richness linked to temperature and deep-sea patterns associated with nitrate-driven organic matter remineralisation.

## Results
### Global patterns of marine biodiversity
The highest number of animal occurrence records was reported in Norwegian waters in shallow (6,258,830) and mesopelagic (809,096), and off California in deep (512,758) seas (Fig. 1). Reported species richness per hexagon peaked in the Gulf of Mexico in shallow (11,419), New Caledonia in mesopelagic (3,809), and northern New Zealand (5,204) in deep habitats. Rarefied ES50 maps revealed unexpectedly high richness in the Arctic and across deep-sea habitats, with notable peaks near Durban (shallow) and the Gulf of Mexico (meso- and deep waters). More than 15% (ca. 81 million km²) of the shallow water, and 32% of the mesopelagic and deep-sea areas (>165 million km²) had

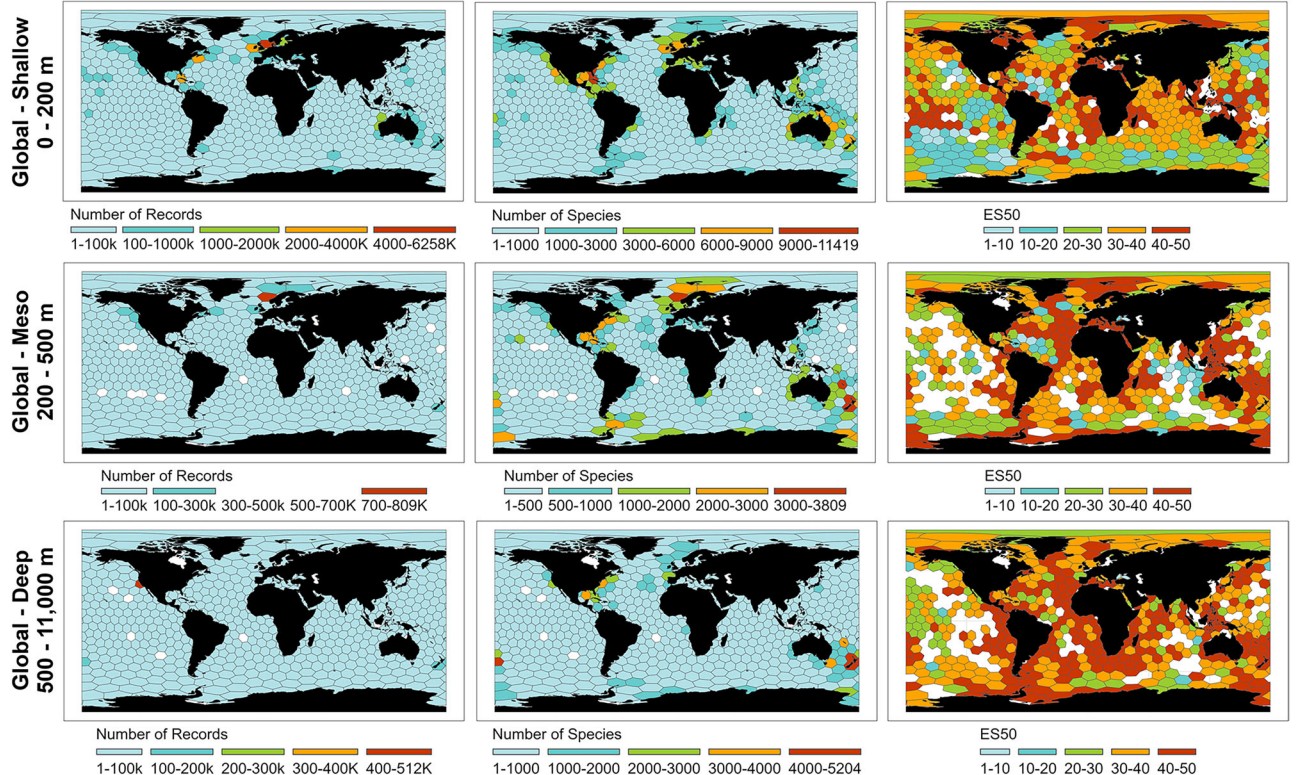

**Fig. 1 | Number of animal occurrence records, species, and expected species (ES50).** The numbers are shown per hexagonal cells (ca. 800,000 km²) and across ocean depths. The white hexagons show areas with no information. For Chao2, Abundance-based Coverage Estimator (ACE), and Inverse Simpson measurements per hexagonal cells across the depths, please see Supplementary Fig. 4.

fewer than 50 marine occurrence records worldwide (per 800,000 km²), especially in the Pacific and Indian Oceans. Chao2 and ACE diversity estimates confirmed the Gulf of Mexico and New Zealand as reported species richness hotspots in shallow and deep waters, respectively (Supplementary Fig. 4). However, species evenness was highest in the western Pacific.

### Taxonomic variation, sampling bias, and data gaps

Sampling effort and biodiversity patterns varied widely across seven major taxa, with Annelida, Chordata, and Arthropoda having the most occurrence records, especially in the North Sea and Northeast Pacific (Fig. 2 and Supplementary Fig. 5). Species richness in Annelida, Arthropoda, and Cnidaria peaked in coastal zones of the North Atlantic and Australian Southwest Pacific, and in Chordata, Echinodermata, Mollusca, and Porifera, peaked in the Northwest Pacific and Caribbean Sea. Rarefied ES50 analyses revealed the Arctic as a richness hotspot for annelids and highlighted the western Pacific as a key area for expected richness and evenness in other taxa, though large data gaps remain across much of the ocean.

Of the ca. 48 million occurrence records across 184,141 marine species, shallow waters contained the highest number of occurrence records (37.94 million) and species counts (131,980) (Supplementary Table 2). The mesopelagic zone and deep sea had 3.97 and 2.72 million occurrence records, respectively, with around 49,000 to 51,000 species count each. Notably, over 3.35 million records belonging to 51,900 species lacked depth data. Chordata had the most occurrence records (25.77 million) but lower species richness (35,996 species) than Arthropoda and Mollusca (42,759 and 42,445 species, respectively).

Species distributions and richness across 5° latitudinal bands exhibited bimodal patterns with a dip at the equator, although these patterns were not statistically different from unimodal distributions and did not appear in rarefied ES50 plots (Fig. 3 and Supplementary

Table 4). The results demonstrated sampling biases, particularly at the equator and around the African countries, the Polar regions, and the deep sea, suggesting new species discoveries by accelerating sampling/data sharing efforts. The Northern Hemisphere had higher sampling/data sharing efforts compared to the Southern Hemisphere, but not higher species count (Supplementary Fig. 6). Overall, data availability and species richness declined with depth, with a pronounced drop between 8000 and 11,000 m, indicating major knowledge gaps (Supplementary Fig. 7).

All seven taxa had more occurrence data in the Northern Hemisphere (Fig. 4). However, species richness, especially for most taxa except Annelida and partly Chordata, was higher in the Southern Hemisphere according to ES50 estimates. While occurrence and richness showed bimodal latitudinal trends, these were not statistically significant (Supplementary Table 4). Increased sampling in mid-latitudes (40–60°) will not necessarily accelerate species discoveries, but it does in other latitudes (Supplementary Fig. 8). Occurrence records and species counts declined with depth across all taxa. However, ES50 plots revealed that deep-sea biodiversity could be comparable to shallow waters, except for a decline of around 4000 m observed across all taxa (Supplementary Fig. 9).

### Environmental drivers across ocean depths

GLM results showed that in shallow waters, species richness and ES50 were best explained by sea surface temperature and nitrate levels (Fig. 5, Supplementary Tables 5 and 6, and Supplementary Note 1). In the mesopelagic zone, human impact, which might reflect more sampling efforts, was the strongest predictor of species counts, while nitrate best explained ES50, with temperature and intercept models also competitive (Supplementary Tables 7 and 8). In the deep sea, nitrate concentrations were associated best with species richness, while ES50 was best explained by the intercept model (Supplementary

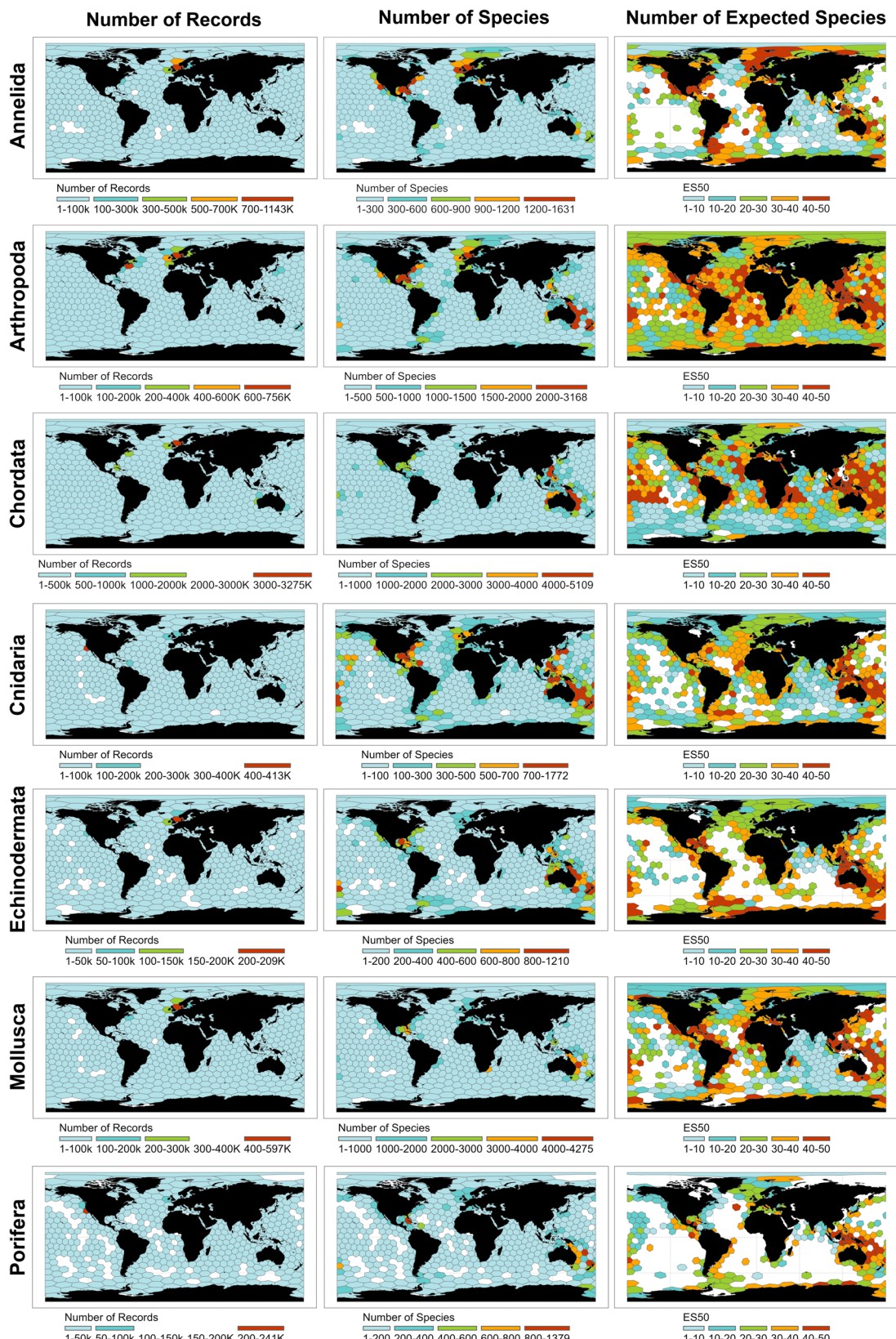

**Fig. 2 | Number of occurrence records, species, and expected species (ES50) for seven taxa.** The numbers are shown per hexagonal cells (ca. 800,000 km²) and across ocean depths. The white hexagons show areas with no information. For Chao2, Abundance-based Coverage Estimator (ACE), and Inverse Simpson measurements per hexagonal cells across the seven taxa, please see Supplementary Fig. 5.

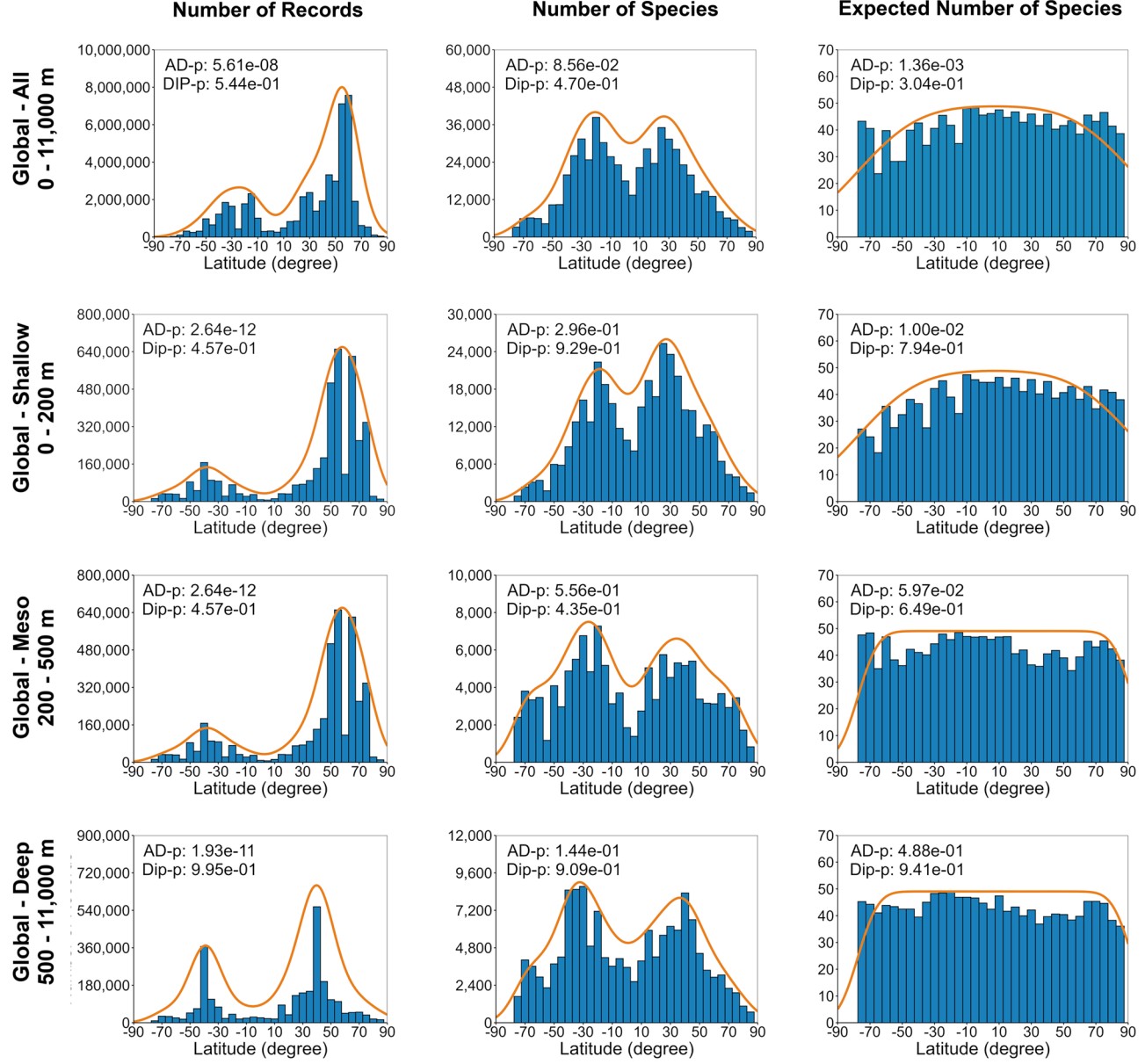

**Fig. 3 | Latitudinal marine animal species richness gradients.** Blue bar plots show the number of occurrence records, observed species, and expected species (ES50) across depths. The orange line shows the kernel density estimation (KDE). Anderson–Darling (AD) and Hartigan's Dip (Dip) tests were used to assess normality and modality, respectively. For bathymetric species richness gradients across depths, please see Supplementary Fig. 7.

Tables 9 and 10). In addition, the number of occurrence records and nitrate models had close competition with the intercept model, with lower support.

GAM results showed that species richness and ES50 were best explained by combined geoecological variables across all depths, except in the mesopelagic zone, where depth was the strongest predictor of species count, and in shallow waters, where ES50 was associated with primary productivity (Supplementary Note 1). Predicted species richness declined at the equator, though this dip disappeared in bias-corrected ES50 models (Fig. 6).

## Discussion

Marine species distributions and biodiversity patterns are a complex interplay of evolutionary adaptations, geological events, sampling and data availability efforts, taxonomic studies, and human influence on the world ocean. This complexity points to the urgent need to document and protect marine biodiversity and ecosystems for

environmental and human well-being. The development of marine biodiversity databases, driven by global data-sharing efforts, established a key advancement in marine science. These databases now follow FAIR (Findable, Accessible, Interoperable, and Reusable) principles[45,46], addressing past issues of fragmented and inaccessible data. With ongoing human impacts on marine life, monitoring and understanding biodiversity changes are crucial to preventing further biodiversity loss.

Despite significant progress in marine biodiversity data digitisation, the global ocean remains vastly undersampled, especially in equatorial and deep waters, with approximately 20% of the world's ocean below 200 meters still lacking data. When sampling biases were corrected, using standardised species richness estimates for a fixed sample size of 50 occurrences (ES50), the diversity of deep-sea ecosystems emerges as much higher than previously reported[47], challenging some earlier assumptions that deep-sea habitats are comparatively less diverse. While many other studies have noted the

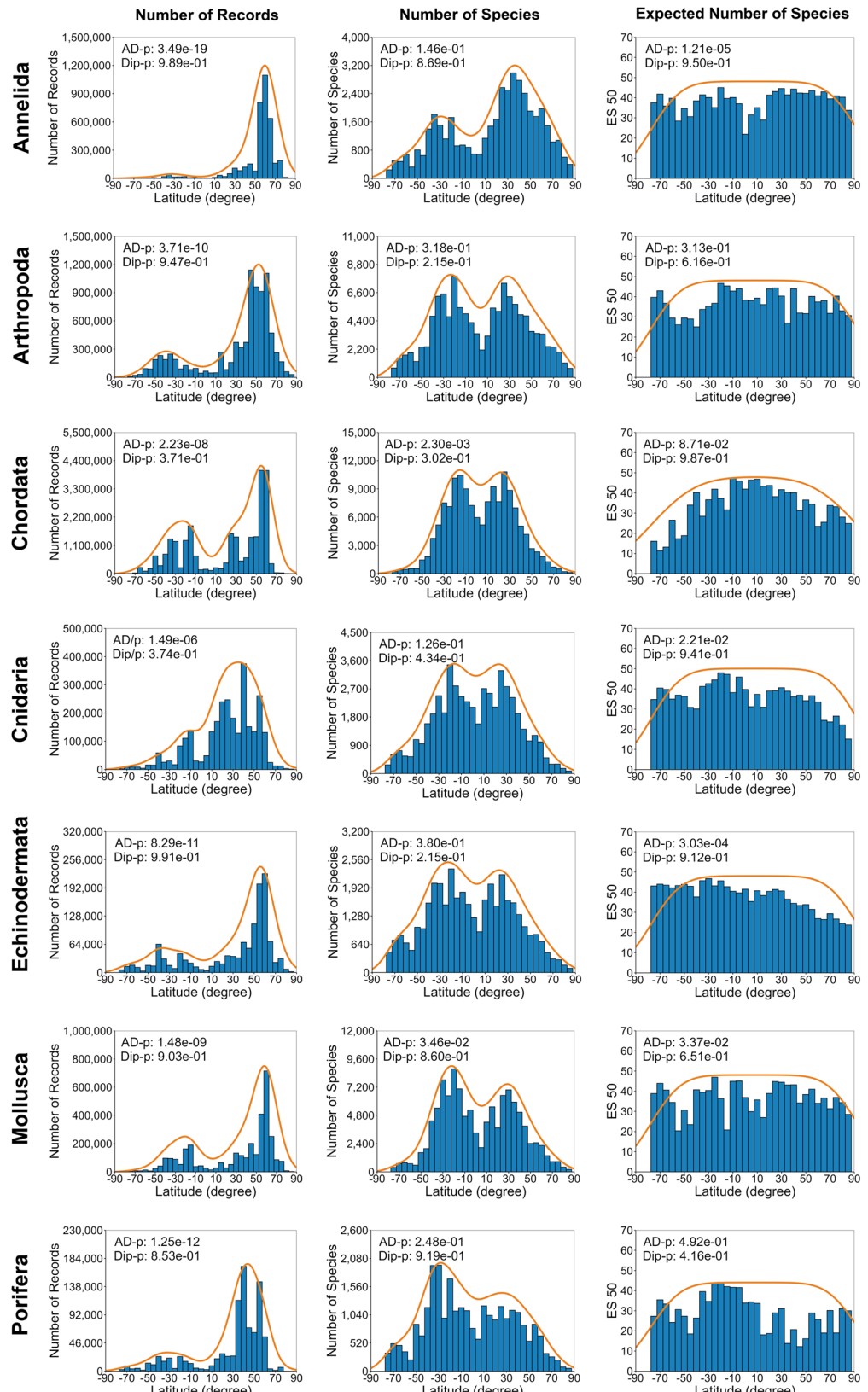

**Fig. 4 | Latitudinal marine animal species richness gradients for seven taxa.** Blue bar plots show the number of occurrence records, observed species, and expected species (ES50) across depths. The orange line shows the kernel density estimation (KDE). Anderson–Darling (AD) and Hartigan's Dip (Dip) tests were used to assess normality and modality, respectively. For bathymetric species richness gradients across taxa, please see Supplementary Fig. 9.

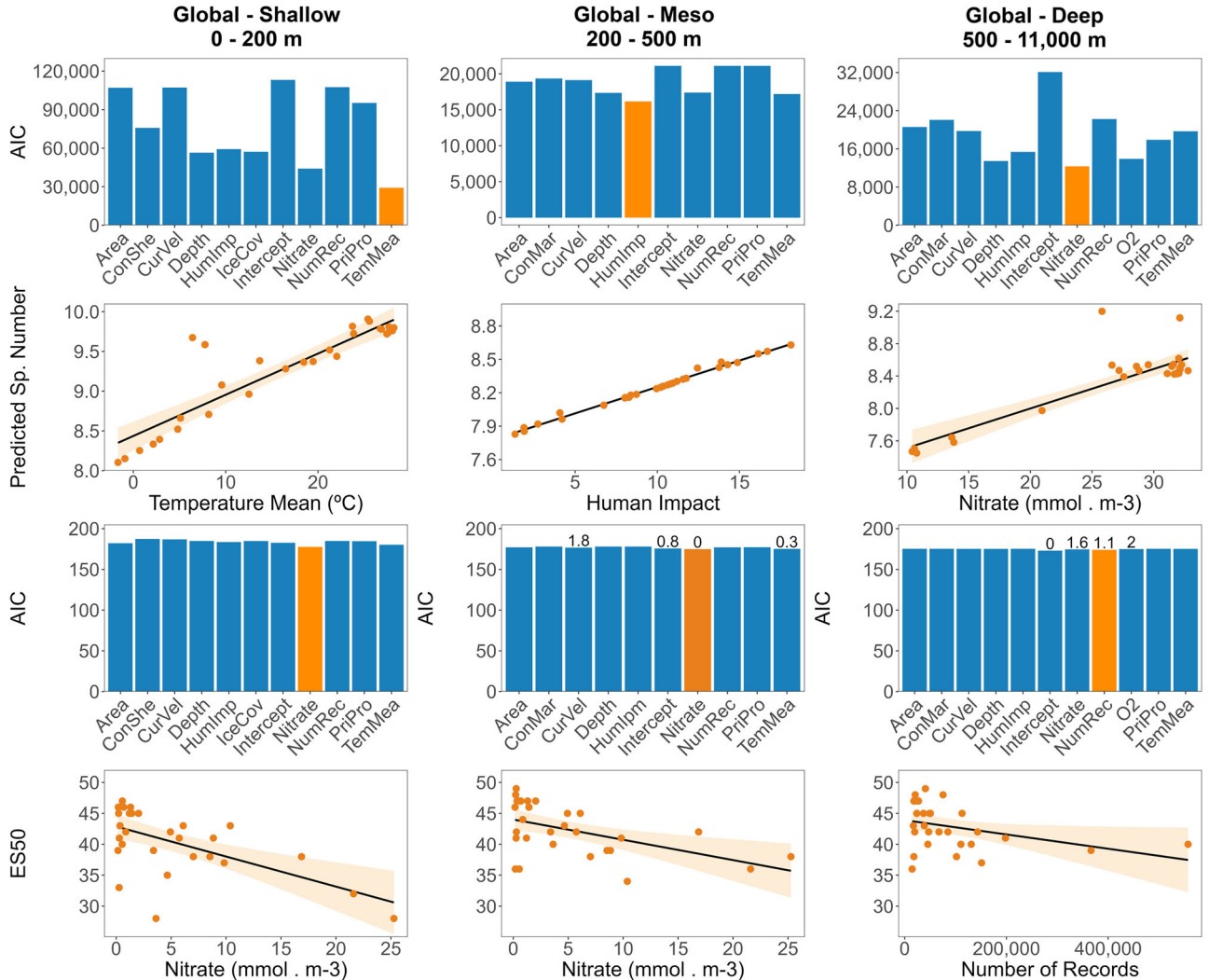

**Fig. 5 | General Linear Model (GLM) outputs of species richness associations with multiple variables.** Bars represent Akaike Information Criterion (AIC) values for each model (blue), with the best-supported models shown in orange, defined as models with the lowest AIC, highest Akaike weights, and ΔAIC ≤ 2 across depth zones. Where multiple variables met the ΔAIC ≤ 2 criterion, values are displayed above the bars to indicate relative model support based on ΔAIC. Scatter plots (orange) illustrate the relationship between the strongest predictor(s) and the predicted response variables for both species' richness and expected species (ES50). The solid black line represents the GLM model prediction, and the shaded orange area indicates the 95% confidence interval around the fitted relationship. Correlation matrices, full response curves, and associated R code are provided in the Supplementary Note 1.

potential for high deep-sea biodiversity, this study reinforces this understanding by providing a quantitative, bias-corrected, and taxonomically broad global assessment that extends previous findings across all ocean depths. This is especially relevant for better-sampled regions such as New Zealand, which has the highest number of species in mesopelagic and deep-sea habitats. This discovery not only highlights the need for more sampling in the deep sea but also suggests that some regions may hold vacant biodiversity hotspot potential that remains largely undocumented. New Zealand exhibited the highest reported deep-sea species richness, likely reflecting concentrated taxonomic deep-sea research efforts in this region. This data deficiency and biased sampling efforts are thus a key limitation in our understanding of marine biodiversity patterns and their drivers, particularly in deep-sea habitats.

This study provides a comprehensive overview of global marine animal knowledge gaps and associated biodiversity patterns, emphasising descriptive insights without implying causal relationships. The results documented that around 50% of marine habitats across all depths have fewer than 50 animal occurrence records in ca. 245 million km² of the ocean area, highlighting the urgent need for more comprehensive sampling efforts across various ocean depths and geographic regions. These gaps are particularly pronounced in equatorial regions, including the Coral Triangle, which is widely recognised as a global biodiversity hotspot[48,49]. The coexistence of high biodiversity potential and low data availability highlights the need for targeted sampling and improved data mobilisation.

Central tropical regions (−5° to 5°) remain persistently underrepresented despite ongoing improvements in data sharing (Supplementary Fig. 10). Although hundreds of thousands of occurrence records have been contributed from these regions over the past decade, they still account for less than 2.5% of global marine occurrence data in major repositories such as OBIS. The poor tropical sampling is reported not only in marine realms but also in terrestrial habitats[4,37,50], indicating a broader influence of sampling bias on global biodiversity assessments. While data availability continues to improve, increased efforts are needed to prioritise data mobilisation in these regions.

In contrast, developed regions, particularly the North Atlantic, around Europe and North America, remain disproportionately well sampled. Most marine biodiversity data continue to originate from those regions, resulting in spatial biases that shape observed global

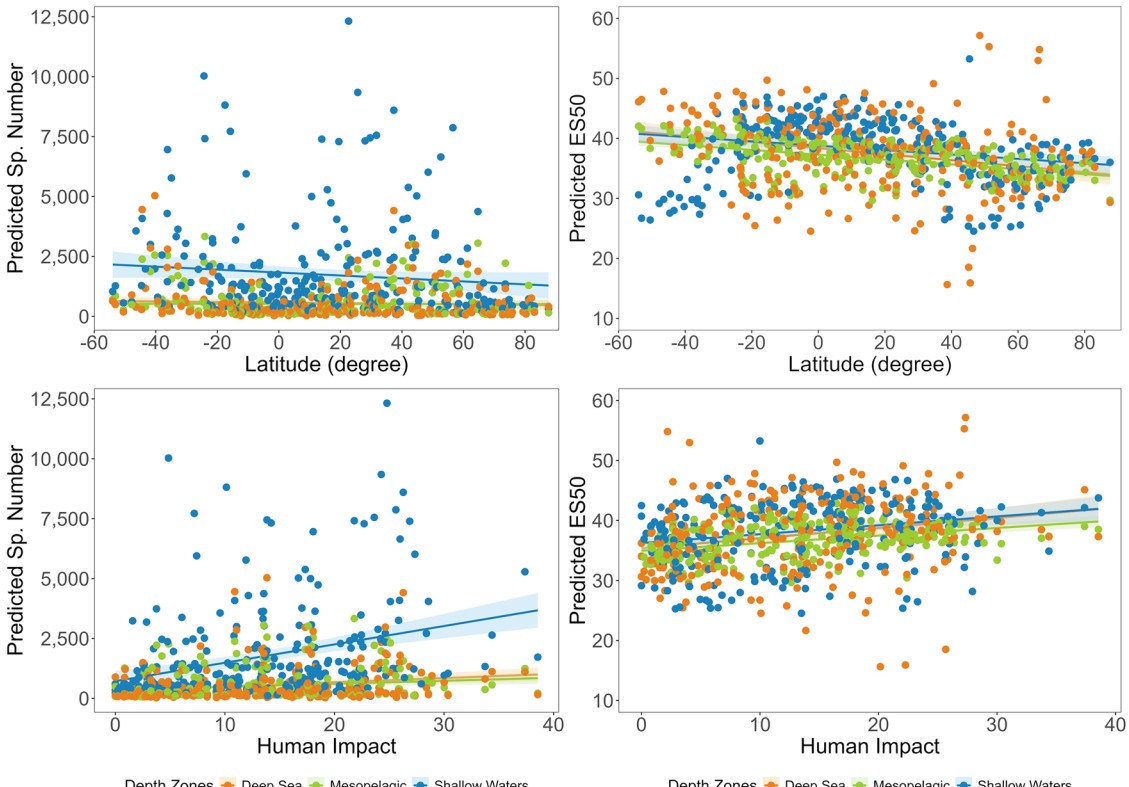

**Fig. 6 | General Additive Model (GAM) outputs of species richness associations with Latitude and Human Impact.** The predicted models for animal species responses to environmental variables across latitude and the Human Impact Index were identified using GAM analysis across 642 hexagonal cells for the predicted number of species and expected species (ES50). The solid lines represent the GAM model prediction, and the shaded areas indicate the 95% confidence interval around the fitted relationship. The correlation matrix, response curves, and corresponding R code are provided in the Supplementary Note 1.

patterns, including perceived gradients in species richness, such as low species richness in the deep sea[47], and debated hypotheses such as bimodal latitudinal species richness gradients[48,49].

Recent global-scale analyses have highlighted a bimodal distribution of marine species richness along latitudinal gradients, peaking at low to mid latitudes but declining at the equator, and a general decrease in species richness with increasing depth across various taxa[31,32,51]. These findings proposed that tropical species evolve in response to temperature fluctuations near the tropics' edges and the availability of highly productive habitats. Some researchers hypothesised ephemeral ecological speciation as a unifying mechanism to explain latitudinal biodiversity patterns[52]. They suggest that ecological speciation occurs more rapidly at high latitudes but that these newly formed species are highly susceptible to extinction through fusion, influencing species turnover, and net biodiversity accumulation[52]. However, these broad patterns may obscure regional and taxonomical variations or habitat types that contribute to understanding the underlying causes[35,53,54]. Some scholars emphasised the need to critically assess Chaudhary et al.'s interpretation of global species richness bimodality, arguing that it may oversimplify complex taxonomic patterns, rely on latitude as an indirect proxy, and reflect biases from uneven sampling and data quality[35]. Later, other scientists estimated that the mid-latitudes are the most sampled regions in the world, and the observed dip in species richness at the equator and polar regions is largely attributed to sampling efforts[37]. The results presented here are consistent with this latter view and extend it by demonstrating that sampling biases affect observed richness patterns across all ocean depths. Analyses based on rarefied species richness estimates indicate that apparent bimodality is not statistically significant, suggesting that previously reported patterns may largely reflect data limitations rather than underlying ecological processes.

By analysing biodiversity patterns across both taxonomic groups and depth zones, this study accounts for variation associated with habitat and taxon-specific responses. Previous studies suggested that not all taxa follow a bimodal richness gradient with latitude, even among the families of a taxon[53,55,56]. Thus, overall patterns may result from taxon-specific variations, habitat types (e.g., coastal, intertidal, continental shelves, deep sea), ecological classification (e.g., benthic or pelagic), sampling and taxonomy bias, data availability, and analysis methods[35,37,53,54,57]. Furthermore, the drivers of species richness vary among shallow-water and deep-sea habitats[15,54,58]. The sampling corrections results here show that deep-sea habitats, once thought to be less diverse, have significantly higher expected species richness than previously reported, supporting other claims[59–61]. Acknowledging data knowledge gap analyses highlights the impact of sampling and data biases on our understanding of marine species distributions and diversity. At the same time, environmental and geological factors remain important determinants of species distributions and richness.

Marine species richness is influenced by a complex interaction of geoecological factors, encountering various spatial and temporal scales[62]. Environmental variables such as temperature, dissolved oxygen, primary productivity, and available nutrients play crucial roles in shaping marine biodiversity trends[31,34,63–66]. Oceanographic variables, including currents, tide heights, and upwelling zones, influence species dispersal and connectivity between habitats, shaping the species distribution patterns[67–71]. Habitat heterogeneity and diversity, such as coral reefs, mangroves, and seagrass beds, accelerate genetic diversity[40,55,72]. Historical events, such as glaciations and past climatic fluctuations, have left imprints on marine biodiversity by driving speciation, dispersal, and extinction rates[65,73–76]. Additionally, planktonic duration increases with latitude. While tropical species have the shortest planktonic durations, observed dispersal distances were

greater in the tropics and at high latitudes, and lowest at mid-latitudes. Biological interactions such as competition, predation, and mutualism delimit the species distribution ranges[77–80]. In addition, human-induced factors like pollution, overfishing, biological invasions, and habitat destruction triggered biodiversity loss in marine ecosystems[81–84]. The current patterns of marine species richness observed in ecosystems are thus the outcomes of interaction between various abiotic and biotic factors[6]. While these broad geoecological factors shape marine biodiversity globally, their influence varies considerably across taxa and depth gradients, driven by taxa-specific characteristics and distinct environmental and ecological dynamics.

The drivers of species richness vary between shallow water and the deep sea due to differences in environmental conditions, habitat types, and ecological interactions. Various factors such as temperature, dissolved oxygen, primary productivity, and habitat shape species richness in shallow water ecosystems[18,72,85]. In contrast, the deep sea is characterised by extreme environmental conditions, including high pressure, low temperatures, and limited light penetration, challenging deep-sea life[15,86]. Despite these harsh conditions, the deep sea harbours a remarkable diversity of species adapted to its specific environment, such as inhabitants of submarine canyons and seamounts[87,88]. Important ecological drivers of deep-sea species richness also include wood-falls, hydrothermal vents, cold seeps, and other chemosynthetic environments, which provide energy sources and create localised hotspots of biodiversity[41,42,89].

Here, ecological analysis reveals that sea surface temperature in shallow waters and nitrate levels in the deep sea are associated with observed gamma species richness. In contrast, human influence was associated with reported species richness in mesopelagic zones, likely reflecting increased sampling effort in more accessible or frequently studied regions. Nitrate in the deep sea likely serves as a proxy for water-mass characteristics and remineralisation processes, rather than a direct driver of species richness[39]. Due to the slow circulation of deep waters, inorganic nutrients such as nitrate accumulate at depth as a result of organic matter decomposition, whereas they are rapidly utilised and depleted in surface waters[39,90]. Accordingly, Fig. 5 showed a negative relationship between nitrate and species richness in shallow and mesopelagic layers, but a positive relationship in the deep sea, highlighting a shift in the underlying ecological processes across depth gradients.

Patterns of Alpha species richness were best explained by a combination of all ecogeographical variables; however, primary productivity overtook this role in shallow waters when sampling biases were accounted for (ES50 species richness). Several results were consistent with the study by Tittensor et al. (2010), such as species richness being correlated with human activities (here only in the mesopelagic fauna) and temperature being a key player (here only in shallow water, but not in the mesopelagic and deep-sea habitats)[17]. Compared to Tittensor et al. (2010), this study compiled a vastly expanded dataset (184,141 species compared to 11,567), applied depth-classified and bias-corrected richness estimates (ES50), calculated both abundance-based richness estimators (ACE and Inverse Simpson) and presence–absence–based estimates (Chao2), and used more advanced modelling (GLMs and GAMs) incorporating additional ecological variables such as nitrate, current velocity, and continental shelf and margin characteristics. These enhancements are critical, as species richness associations vary markedly across ocean depths and must be evaluated within a bathymetrically structured framework.

Overall, sampling biases and unequal data availability remain major constraints on understanding global marine biodiversity. Approximately half of the ocean still lacks adequate data coverage, with particularly large gaps in deep-sea and equatorial regions, despite recent advances in data sharing. Remarkably, 20% of the ocean below 200 meters is still undersampled, and a large portion of the data gathered is not publicly available. Although deep-sea undersampling is

well recognised, analyses provide a quantitative, spatially explicit estimate of this gap, rather than introducing it. This persistent deficit distorts our understanding of deep-sea biodiversity and highlights the urgent need to close spatial data gaps through expanded sampling and improved data sharing. Accurately describing the patterns of marine biodiversity and the factors that influence them requires this knowledge.

Persistent data gaps in recognised biodiversity hotspots such as the Coral Triangle further emphasise the need for improved data sharing and targeted sampling. Differences in environmental drivers across depth zones, such as temperature in shallow waters and nutrient availability in deep seas, highlight the importance of depth-explicit analyses for biodiversity monitoring and conservation. Nevertheless, these associations must be interpreted with caution because variables such as temperature and latitude covary strongly, limiting our ability to infer causation.

In conclusion, substantial portions of the global ocean remain inadequately explored or underrepresented in shared data, limiting accurate estimates of global marine biodiversity. Addressing these gaps will require coordinated international efforts to expand sampling coverage, improve data accessibility, integrate Essential Ocean Variables (EOVs)[91], and strengthen long-term monitoring of ocean life. A more complete understanding of marine biodiversity patterns and their underlying drivers is essential for informing conservation strategies and sustainable ocean management under ongoing environmental change.

## Methods

### Biodiversity data collection and quality control

The study area covers all oceans and enclosed seas worldwide from −90°S to 90°N, and −180°W to 180°E, from 0 m to 11,000 m. Base polygons of the global oceans and seas were initially extracted from the Marine Regions[92]. The extracted ocean polygons did not cover the whole ocean area. Thus, they were modified to ensure complete and accurate coverage of all ocean and enclosed sea boundaries for extracting species occurrence data. Modifications in QGIS 3.36.2 included adjusting boundaries, merging or splitting polygons where necessary, and correcting gaps or overlaps to fully represent the study area from −90°S to 90°N and −180°W to 180°E (Supplementary Fig. 1). Well-Known Text (WKT) was then generated for each generated ocean polygon to extract occurrence records of Animalia from OBIS and GBIF using packages such as "robis" and "obistools"[93] and rgbif[94]. The statistical software R 4.3.2 was used for data extraction, management, quality control, analysis, and visualisation. Non-animal kingdoms, including Bacteria, Chromista, Plantae, Protozoa, Fungi, or Viruses, were outside this study's scope. This scope limitation reflects current data availability and the already large scale of the analysis. Readers should note that some biodiversity patterns, particularly those of microbial and non-animal taxa, are not captured here.

To clean and quality-control the merged dataset following OBIS Manual and[66,95], all duplicate records and those meeting the following criteria were removed using the R package obistools[93]: absent data, fossil records, doubtful or missing coordinates, occurrences mapped on land, or if the reported depth exceeded the maximum bathymetry for that location, with a depth margin threshold set at 50 m[95], and those with coordinate uncertainty exceeding 100 km. Species names were taxon-matched against the World Register of Marine Species (WoRMS) to reconcile synonyms and misspellings. Only accepted marine and brackish water species (Animalia) were retained for analysis. A final quality-controlled dataset of 47,995,228 occurrence records (citations to the dataset are in Supplementary Table 1 and Supplementary Data 1) belonging to 184,141 accepted marine species was compiled for the biodiversity analyses (Supplementary Fig. 2).

All occurrence records were classified into three depth groups: shallow-water (0–200 m), mesopelagic (200–500 m), and deep-sea

(500–11,000 m)[96,97], based on the reported depth for each record (Supplementary Fig. 3). A total number of 3,359,550 occurrence records lacking depth information were excluded from depth-specific analyses (Supplementary Table 2). For taxon-specific analyses, seven dominant taxa, including Annelida, Arthropoda, Chordata, Cnidaria, Echinodermata, Mollusca, and Porifera, were selected, representing ca. 76% of occurrence records and ca. 93% of species in OBIS and GBIF, while the remaining phyla were excluded due to their limited data availability.

### Geoecological data collection and generation

Ecological and bathymetry variables were extracted from Bio-ORACLE including sea water/current velocity (m.s-1), sea ice cover (Fraction), photosynthesis available/active radiation (E.m-2.day-1), primary productivity (mmol. m-3), dissolved molecular oxygen (mmol. m-3), nitrate (mmol. m-3), ocean temperature (°C), and bathymetry (m). For all ecological variables, mean, maximum, and minimum values were obtained for both pelagic layers (shallow water and mesopelagic) and benthic layers (deep-sea)[98,99]. The continental shelves were derived from the bathymetry raster layer by selecting all grid cells with depths from 0 to ≤ 140 m. The continental margins (between > 140 m and 3500 m depth) were initially extracted from the Marine Regions[43] and transformed into a raster layer in QGIS for data extraction based on 5° latitudinal bands. The Global Human Influence Index (HII) layer was extracted from NASA Earth Data[44]. The HII layer is a 1-km resolution global dataset created from nine global data layers covering human population pressure (population density), human land use and infrastructure (built-up areas, nighttime lights, land use/land cover), and human access (coastlines, roads, railroads, navigable rivers), developed by the Wildlife Conservation Society and CIESIN.

### Biodiversity data and gap analyses

To standardise sampling effort and reduce biases, animal species occurrences were aggregated into equal-sized hexagonal grids (ca. 800,000 km²), 5° latitudinal bands, and equal depth intervals. Using hexagonal grids as the unit of analysis minimises edge effects, smooths over local variability, and accounts for uneven sampling across regions and depths[36,86]. The chosen spatial resolutions of ca. 800,000 km² hexagonal grids balance global coverage, sampling completeness, and model reliability (reducing the number of grids with zero occurrences), while also improving map visualisation and colour distinction. However, this coarse scale may obscure finer local patterns, such as coastal reefs or seamount hotspots, and this limitation should be acknowledged. Additionally, 5° latitudinal bands were used because this resolution is widely used in global marine biodiversity research[17,29,47] and offers a practical balance between ecological relevance and statistical robustness, providing adequate sample size per band while minimising spatial bias from uneven sampling across latitudes. Six measures of species richness were calculated for each hexagonal cell to compare species richness. These measures included: 1) the number of occurrence records, 2) the number of unique species, 3) expected species richness per 50 individuals (ES50), 4) Chao2, 5) Abundance-based Coverage Estimator (ACE), and 6) Inverse Simpson using the 'vegan' R package[100]. The ES50 method[100] was used to estimate species richness, where ES50 represents the expected number of species in a random sample of 50 individuals in both hexagonal cells and 5° latitudinal bands. The extrapolation is performed using individual-based rarefaction: a rarefaction curve is generated from the observed species individuals, and the expected number of species for a standardised sample of 50 individuals is calculated. For each species, the probability of being represented at least once in a random subsample of 50 individuals is computed, and these probabilities are summed across all species. This approach provides a more accurate estimate of species richness while controlling for uneven sampling effort. Chao2 species richness was estimated from presence–absence data using the specpool() function, providing an asymptotic estimate of total species richness that accounts for rare species based on the number of species occurring in only one or two samples. Species richness was also estimated using the ACE via the estimateR() function, which provides an asymptotic richness estimate emphasising rare species based on their individual numbers, rather than solely on presence–absence data. The Inverse Simpson index using the diversity() function with index = "invsimpson" was also estimated, accounting for both species richness and evenness while giving greater weight to abundant species.

Kernel density estimation (KDE) was applied using a Gaussian kernel to visualise uni-, bi-, or multi-modal patterns, overlaid on histograms of 5° latitude bins and 10 m depth bins. The bandwidth was set to 10° for latitude and 20 m for depth to balance smoothing while preserving patterns in regions with limited sampling efforts. The Anderson-Darling (AD) test using the nortest R package[101] was used to evaluate normality, while the Dip test using the diptest R package[102] assessed the modality of the distribution. Data processing and visualisation were conducted using the R packages: readxl, dplyr, tidyverse[103,104] and ggplot2[105].

### Spatial modelling of species Count and ES50

Since aggregating hexagon-based data into 5° latitudinal bands significantly reduced sample size and constrained our ability to fit complex functional responses, both Generalised Linear Models (GLMs) (per 5° latitudinal bands) and Generalised Additive Models (GAMs) (per hexagonal cells) were applied to assess how geoecological predictors influence species count and ES50[86,106,107]. GLMs, applied to gamma species richness (species counts per 5° latitudinal bands), enabled robust estimation of linear relationships while accounting for variation in sampling effort via the number of records per band. GAMs, applied to alpha species richness (species counts per hexagon), captured complex, non-linear responses and finer spatial patterns, including smooth terms for environmental gradients and spatial autocorrelation. The combination of GLM and GAM ensures the detection of simple linear effects at broad scales and more complex spatially structured effects at finer scales.

Before modelling, collinearity among predictors was assessed using pairwise correlations (Supplementary Note 1). Dissolved oxygen ($O_2$) showed a strong correlation with mean temperature in both shallow-water and mesopelagic habitats (a correlation coefficient of 1), indicating potential multicollinearity (Supplementary Fig. 11). To reduce this issue and improve model interpretability, $O_2$ was excluded from the analyses for these habitats. In contrast, this strong correlation was not observed in the deep-sea environment; therefore, $O_2$ was retained as a predictor in the deep-sea analyses. The geoecological variables included in the GLM and GAM were the total area of continental shelves (for shallow waters), continental margins (for mesopelagic and deep-sea regions), ocean area, average human impact, maximum depth, mean temperature, current velocity, ice cover (for shallow waters), nitrate, primary productivity, and $O_2$ (for deep-sea regions).

The GLM models were fitted using a Poisson error distribution, with the number of records per band incorporated to account for variations in sampling effort. Given the presence of zeros in the species count data, a negative binomial error distribution was used for the GAM model. Models were fitted via restricted maximum likelihood, with automatic predictor selection from the mgcv R package[108] to regulate the complexity of smooth terms. An intercept-only model was set as the null hypothesis, assuming that the response variables were not influenced by environmental factors, spatial sampling bias, or spatial autocorrelation, and modelling species counts and the total number of records per locality (sampling effort). However, models using ES50 as the response variable excluded record count as an explanatory variable, as ES50 calculations already account for sampling effort. To account for spatial autocorrelation in predictor and response variables, a two-dimensional spherical spline based on the

latitude and longitude of sampling sites was used. The final models were evaluated using the Akaike Information Criterion (AIC); lower AIC scores provide a better trade-off between accuracy and complexity. A difference in AIC (ΔAIC) of less than two was deemed inconclusive when comparing models.

## Reporting summary
Further information on research design is available in the Nature Portfolio Reporting Summary linked to this article.

## Data availability
The compiled and processed data used in this study are publicly available at the Ocean Biodiversity Information System and the Global Biodiversity Information Facility databases. Citations, including digital object identifiers for the datasets retrieved from the Global Biodiversity Information Facility, are provided in Supplementary Table 1, while dataset identifiers and corresponding citations for records obtained from the Ocean Biodiversity Information System are listed in Supplementary Data 1.

## Code availability
All the R codes are publicly available on GitHub https://github.com/haniehsaeedi/Gaps-and-drivers-of-global-marine-animal-biodiversity-from-surface-to-Abyss/tree/main and Zenodo https://zenodo.org/records/19629992 [https://doi.org/10.5281/zenodo.19629991]. The markdowns of the GLM and GAM analyses are in the Supplementary Note 1 of the Supplementary Information file.

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

## Acknowledgements

I sincerely thank all the data providers and managers from OBIS and GBIF for their contributions, which made this study possible. I especially thank Angelika Brandt, who has always been an inspiring mentor, encouraging me in writing this paper and providing feedback on this work before submission. I also sincerely thank James Reimer and Adnan Shahdadi for their insights. I am very grateful to Keyvan Allahyari for refining the English language of this manuscript.

## Author contributions

H.S.: conceptualisation, methodology, data collection, data quality control, data analysis, creating the graphs and tables, writing the manuscript.

## Funding

## Competing interests

The authors declare no competing interests.
