## [Peer Review file · Nature Communications]

Gaps and drivers of global marine animal biodiversity from the surface to abyss

Corresponding Author: Dr Hanieh Saeedi

Version 0:

Reviewer comments:

Reviewer #1

(Remarks to the Author)

“Gaps and drivers of global marine biodiversity from surface to Abyss”

The manuscript addresses a fundamental and timely question: Where are the global marine biodiversity hotspots and knowledge gaps from the ocean surface to the abyss, and what ecological drivers underlie these patterns? This research question is highly relevant in the context of current marine science and conservation. Previous studies have long debated marine biodiversity gradients, often focusing on latitude (with recent work suggesting a bimodal richness pattern peaking at mid-latitudes rather than the equator) and on specific depth zones or taxa. However, a comprehensive synthesis across all ocean depths and multiple taxonomic groups has been lacking.

Main concerns

That said and despite the large number of data analysed there are some inherent problems of the present ms:

1. the observational nature of the study does not allow to infer on causalities (cause-effect relationships);
2. some conclusions are dictated by the data availability and for most (>50%) of the blue Planet data are insufficient to draw any conclusion, nor even to formulate realistic hypotheses.
3. Some conclusions are completely untrue (even when suggested by the data set analysed): New Zealand has the highest number of species in the mesopelagic and deep-sea habitats.
4. Similarly the conclusion that: “The gamma species richness ...e in the deep sea, it is primarily influenced by nitrate concentrations” is ecologically not credible as nitrates are not drivers, rather possibly descriptors of the re-mineralization process.
5. The gamma species richness in shallow waters is predicted to be driven by sea surface temperature... this conclusion has been previously reported and published in Nature But the main problem I have with the ms is its originality. The ms by Tittensor, D., Mora, C., Jetz, W. et al. (2010) entitled: “lobal patterns and predictors of marine biodiversity across taxa”. Nature 466, 1098–1101, stated: “Spatial regression analyses revealed sea surface temperature as the only environmental predictor highly related to diversity across all 13 taxa.”.
6. the conclusion that the deep sea is undersampled is not novel and largely agreed at international level.
7. The authors suggest that the previously described bimodal patterns are likely an artifact of lower sampling effort in equatorial regions, but this conclusion has been repeatedly presented by Fernández and Marques (2017) and Menegotto & Rangel (2018) among the others (I’m teaching that at my marine biology courses...). Thus the authors cannot claim this as a novel finding.
8. A key finding of the present study is that deep-sea habitats are much more diverse than previously reported once sampling bias is accounted for. But again this is part of a consolidated knowledge.
9. In addition, the study focuses on Animalia (and primarily on several major phyla). Other components of marine biodiversity – such as protists, fungi, flora (algae, seagrasses), and prokaryotes – are not included. This is understandable given data availability and the already enormous scope, but it means the “global marine biodiversity” assessed is not truly the entirety of biodiversity (it’s essentially the multicellular animal biodiversity). Some patterns (especially for microbial diversity, which can differ) are not captured. The authors might consider noting this as a scope limitation for clarity.
10. The Human Impact Index correlating with mesopelagic diversity might only partly reflect that regions, with high human presence (e.g., North Atlantic) were better sampled rather than genuinely higher in intrinsic diversity. The models did include sampling effort as a covariate in GLMs, which might help, but interpreting human impact causally is tricky.

11. Similarly, temperature and productivity can covary with latitude. The authors have tried to disentangle these with their models (and by separating depth strata), but caution is warranted in attributing causation.

Thus, my issue here is that I do not see take home messages that are new nor producing a real improvement of the present knowledge.

Other concerns:

1. The spatial and categorical structuring of the data classified available information into three depth zones – shallow (0–200 m), mesopelagic (200–500 m), and deep-sea (500–11,000 m). But mesopelagic is deep and the classification presented is thus misleading. In addition aggregating data in this way means considering 5%-8% and 87% which makes extremely unbalanced the assessment in terms of robustness of results.

2. Seven major taxa (Annelida, Arthropoda, Chordata, Cnidaria, Echinodermata, Mollusca, Porifera) were analyzed separately in addition to the overall dataset. The choice of focusing on these dominant animal phyla is understandable yet questionable as there are 35 marine phyla (here only 7 considered).

3. The use of ~800,000 km² hexagonal cells and 5° latitude bands is pragmatic for a global view, but it is a coarse spatial grain. Finer-scale patterns (e.g., coastal reef hotspots or seamount clusters) might be obscured when averaged over such large areas. For instance, the biodiverse Coral Triangle region might not register as a hotspot in this analysis because it is under-sampled and because any given hexagon there may include more open ocean than reef. Thus, some nuance is lost at local scales. The authors could perhaps discuss this scale issue, though it does not invalidate the broad findings.

In conclusion, there are merits in the present work, especially in quantifying Global Data Gaps (we know there are gaps but we need better estimates), yet this result is not sufficient to guarantee the publication in a high-ranking journal as Nature Communications.

(Remarks on code availability)

Reviewer #2

(Remarks to the Author)

The Author presents a very interesting paper on sampling biases in biodiversity data of marine species across all the ocean column. After compiling and curating a database of marine species occurrences, the Author calculated the number of occurrence records, the number of unique species, the expected species richness per 50 individuals, the Chao2 metric, the abundance-based coverage estimator, and the inverse Simpson metric. The results show that there are many important gaps and biases in the distribution of marine species data. The paper is well written, the methods are correctly implemented, the results are sound, and the conclusions are based on the results. However, there are some issues that should be clarified. In consequence, I cannot recommend the publication of the manuscript in the present state.

L14: Species hotspots are not clear enough. It should refer to species distribution hotspots.

L21: Please indicate how the data were modelled. The abstract lacks information about the methods. It also lacks information about what proportion of the total biodiversity the 184 thousand species represent.

L124: Why were the ocean polygons modified? What modifications were applied?

L134: What is the accuracy of the species depth? Is it 50 m higher than that accuracy? In other words, what was 50 m selected as the threshold?

L137: It should make clear across the manuscript (even in the title) that the analyses only include animals, not all marine groups.

L144: How many records do not have depth information?

L152-153: Please, rephrase. Do you mean that the mean, maximum, and minimum values were obtained for all variables?

L155: I do not understand what you mean by 5-degree latitudinal bands. Why bands of 5 degrees? Please provide a justification.

L156: It is not clear to me how the continental shelves were obtained.

L162: What is the size of the hexagon? 5-degrees? It is not clear.

L66: Please, explain in detail the 'expected species richness per 50 individuals'. It should be indicated at least that ES50 is the expected number of marine species in a random sample of 50 individuals. Please also explain how the extrapolation is performed.

L166-168: Similarly to the previous point, please define the Chao2, the Abundance-based Coverage Estimator, and the Inverse Simpson metrics. Not all readers will be familiar with these methods.

L173: Please, provide the parameters for the kernel estimator.

L180: Why are GLM applied to 5-degree bands and GAM to hexagonal cells? Please provide a justification.

It would be useful for a better interpretation of the results to have a map with the limits of the depth zones. This will help to interpret the maps in Figure 2. Cnidaria should be more restricted to shallow zones.

L241: The highest number of occurrence records and species counts in shallow waters is not reflected in Figure 1.

Figure 3: I suggest indicating the y-axis as a subtitle of each column of plots. The y-axis can be renamed to frequency.

Supplementary Figure 2: The size of the points should be reduced. This will make the gaps in the point occurrences clearer.

(Remarks on code availability)

The Author provided the code as a PDF, not as an R file. However, the users can copy the text from the PDF and extract the code.

Version 1:

Reviewer comments:

Reviewer #2

(Remarks to the Author)

I am very pleased with how the author has incorporated all my suggestions. The manuscript has been significantly improved. I have no further adjustments to suggest.

(Remarks on code availability)

Reviewer #3

(Remarks to the Author)

This is a novel study that provides a robust suite of analytical/methodologic tools to assess the most recent available data and biodiversity patterns. The clear, well-organized analytical framework presented will help advance to research that aims to dive deeper into the causality of biodiversity, and as is presents a strong backbone in which to reference for data-gaps that may be distorting future researchers' interpretations. This MS offers a good source for future analyses to use & cite – as it incorporates the bias and updated records of occurrences.

Minor comments:

- Variables in the models not well explained in the text introduction
- Figure 5 Fix typo in Temperature reference. In the plot "TheMean" and "Themperture" in correlation plots. Additionally, the geo-ecological data variables should be written out either on the plots or in the captions.
- Figure 5- why is the nitrate relationship positive in the deep sea (i.e. increased nitrate with increase species) vs negative in the shallow and mesopelagic?
- A more in depth definition of the Global human influence index is needed, linking to NASA Earth Data is insufficient for readers that have no familiarity with this. I also think this should be addressed early on in the MS.
- Please make explicit how the sampling bias is accounted for here, referenced in Figure 6, and throughout the manuscript. Helpful to link to the ES50 analysis.
- Human impact should be defined in the main text, what does this consist of? Meaning?
- The added explanation of the metrics used (R2; i.e., Chao2, ES50) is helpful in presenting this to the broad audience of Nat Com.
- line 443-446 should be repeated in the intro as the variables assessed here.
- Line 446- 447 can you clarify why oxygen was not included in shallow and mesopelagic models?

Minor question (more of a curiosity): This storyline details how we have more data than ever before but it's still not enough to draw definitive conclusions. I'm curious what addition of occurrence data in the last decade has been from the tropics/ and/or the global south, are we getting better with our efforts at all? Or are the majority of increases in data collection still from the same data-rich locations? Given the extreme bias in marine biodiversity data, with equatorial areas most understudied, would it be prudent to show trends of where/when these more recent contributions occurred (perhaps in the supplement), linking to policy and references to solving the problem?

(Remarks on code availability)

RESPONSE TO REVIEWERS

I would like to thank both reviewers for their constructive and thoughtful feedback. I appreciate their time and valuable thoughts. Their comments have helped substantially improve the clarity, structure, and quality of the manuscript. Below, a detailed, point-by-point response to both reviewers is provided. All line numbers refer to the revised manuscript by the track change option. Reviewer comments are shown in *italics*, followed by my responses in **bold**.

REVIEWER COMMENTS

Reviewer #1 (Remarks to the Author):

The manuscript addresses a fundamental and timely question: Where are the global marine biodiversity hotspots and knowledge gaps from the ocean surface to the abyss, and what ecological drivers underlie these patterns? This research question is highly relevant in the context of current marine science and conservation. Previous studies have long debated marine biodiversity gradients, often focusing on latitude (with recent work suggesting a bimodal richness pattern peaking at mid-latitudes rather than the equator) and on specific depth zones or taxa. However, a comprehensive synthesis across all ocean depths and multiple taxonomic groups has been lacking.

Response: I am grateful for the reviewer's comprehensive assessment and appreciate the recognition of the study's broad relevance. Addressing the reviewer comments, I have substantially revised the manuscript to clarify the study's unique contribution: the integration of around 48 million standardised marine animal occurrence records across all ocean depths, the quantification of sampling bias and data gaps at a global scale, and the depth-explicit modelling of ecological drivers from epipelagic to hadal zones using harmonised metadata.

Main concerns

That said and despite the large number of data analysed there are some inherent problems of the present ms:

1. The observational nature of the study does not allow to infer on causalities (cause-effect relationships)

Response: Thank you for the comment. Agreed that observational studies cannot establish causal relationships. However, the aim of this paper is not to infer causality but to identify knowledge gaps and map biodiversity patterns across the globe based on the available data. To minimise sampling biases in the comparisons, ES50 and Kernel estimations were calculated, which standardise sampling effort and allow a more robust assessment of patterns across studies. Chao2 was estimated, which is only based on presence and absence data, independent of the sample size. While ES50 improves comparability and reduces bias, it does not change the fact that our analysis is descriptive rather than causal. Addressing your comment, this sentence was added to the Abstract: "This study provides a more robust description of global marine animal knowledge gaps and associated biodiversity patterns while remaining strictly descriptive rather than causal," (Lines 19–21) and "Our study offers a comprehensive overview of global marine animal knowledge gaps and associated biodiversity patterns, emphasising descriptive insights without implying causal relationships." in the Discussion (Lines 442–443).

2. Some conclusions are dictated by the data availability and for most (>50%) of the blue Planet data are insufficient to draw any conclusion, nor even to formulate realistic hypotheses.

Response: This is true, but this is actually the purpose of this knowledge gap analysis: to show the area of high and low occurrence availability, and how these impact our understanding of marine

biodiversity. ES50 was used to standardise sampling effort and to provide a robust description of biodiversity patterns and highlight key knowledge gaps without inferring causal relationships. Using the species response curves from the rarefaction method, the areas where species richness patterns are underestimated and where we expect more species to be discovered, or where the data will be shared, were identified. It is agreed that data sparsity is a key constraint. Our goal was precisely to quantify and map these data gaps, not to infer complete biodiversity patterns globally. It is now revised and clearer in the Discussion, emphasising that the unsampled or data-deficient areas of the ocean remain a major frontier for exploration “Despite the wealth of marine biodiversity data available, vast unsampled or data-deficient regions of the ocean remain a major frontier in our understanding of true global marine life diversity.” (Lines 573–575).

3. Some conclusions are completely untrue (even when suggested by the data set analysed): New Zealand has the highest number of species in the mesopelagic and deep-sea habitats.

Response: Thank you for noting this. You are right, the apparent high reported species richness around New Zealand results from disproportionately high sampling intensity in the mesopelagic and deep-sea habitats. However, even when the highest occurrence records occurred in the Norwegian waters in the mesopelagic area, and off California in deep-sea habitats, the largest number of species was reported from New Zealand, also in the ES50, accounting for sampling biases (Figure 1). This was initially discussed in the Discussion “When sampling biases are corrected, the diversity of deep-sea ecosystems emerges as much higher than previously reported 55, challenging earlier assumptions that deep-sea habitats are comparatively less diverse. This is especially relevant for better-sampled regions such as New Zealand, which has the highest number of species in mesopelagic and deep-sea habitats.” (Lines 428-433).

However, this statement is now cautiously rephrased in the Results by referring to “Reported species richness” not “Species Richness” (Line 304) and Discussion: “New Zealand exhibited the highest reported deep-sea species richness, likely reflecting concentrated taxonomic deep-sea research efforts in this region.” (Lines 436–438). I have also removed this statement from the Highlights of the paper, as it could be misleading based on your comment.

4. Similarly the conclusion that: “The gamma species richness ...e in the deep sea, it is primarily influenced by nitrate concentrations” is ecologically not credible as nitrates are not drivers, rather possibly descriptors of the re-mineralization process.

Response: Thank you very much for this important ecological clarification. The text now was modified to state that nitrate likely acts as a proxy variable reflecting deep-water mass characteristics and remineralisation processes rather than a direct causal driver of richness, in the Discussion: “Deep ocean water circulates slowly, so nitrate and phosphate get stuck deeper in the ocean, even though they are produced from organic matter at shallower depths through remineralisation (Kvale et al., 2019)” (Lines 536–538).

5. The gamma species richness in shallow waters is predicted to be driven by sea surface temperature... this conclusion has been previously reported and published in Nature But the main problem I have with the ms is its originality. The ms by Tittensor, D., Mora, C., Jetz, W. et al. (2010) entitled: “Global patterns and predictors of marine biodiversity across taxa”. Nature 466, 1098–1101, stated: “Spatial regression analyses revealed sea surface temperature as the only environmental predictor highly related to diversity across all 13 taxa.”.

Response: This has already been acknowledged that Tittensor et al. (2010) previously examined the correlation between species diversity and sea surface temperature. Their paper is explicitly cited and compares the generated results with theirs.

However, this manuscript differs substantially from Tittensor et al. (2010) and provides several important advances: (1) used a vastly expanded dataset (>47 million occurrence records and 184,141 species, compared to 11,567 species in 2010); (2) analysed depth-classified and bias-

corrected richness patterns; (3) quantified spatial data gaps; (4) calculated different measures of the species richness based on both abundant (ES50, ACE, and Inverse Simpson) and presence-absence species data (Chao2) to compare how these indices vary globally (5) applied both GLM and GAM modelling approaches, with GAMs specifically capturing the non-linear relationships characteristic of ocean environments; (6) included a broader range of ecological variables, including nitrate, current velocity, and others in our models; and (7) calculated the available ocean area based on depth classes and extracted environmental variables separately for each depth zone. This also addresses the referee concerns of stating that only one environmental driver is associated with all the species richness, which is not true for all taxa and across different ocean depths.

The summary of these comparisons was added to the Discussion to emphasise these novel contributions “Compared with Tittensor et al. (2010), this study compiled a vastly expanded dataset (184,141 species compared to 11,567), applied depth-classified and bias-corrected richness estimates (ES50), calculated both abundance-based richness estimators (ACE and Inverse Simpson) and presence-absence-based estimates (Chao2), and used more advanced modelling (GLMs and GAMs) incorporating additional ecological variables such as nitrate, current velocity, and continental shelf and margin characteristics. These enhancements were necessary because species richness associations vary markedly across ocean depths and must be considered separately.”(Lines 545–551).

6. The conclusion that the deep sea is undersampled is not novel and largely agreed at international level.

Response: I agree that undersampling of the deep sea is well known. But the contribution of this paper is a quantitative global estimate (e.g., ca. 165 million km² below 200 m with <50 records per 800,000 km²), providing spatially explicit evidence of the knowledge gap, to inform the policy domains and to guide future sampling and data sharing efforts.

To address your comment, I have clarified that our findings quantitatively support this rather than reveal this knowledge in the Discussion, “Although deep-sea undersampling is well recognised, our study provides a quantitative and spatially explicit estimate of this gap, supporting rather than introducing this knowledge. This persistent deficit distorts our understanding of deep-sea biodiversity and highlights the urgent need to close spatial data gaps through expanded sampling and improved data sharing.” (Lines 560–563).

7. The authors suggest that the previously described bimodal patterns are likely an artifact of lower sampling effort in equatorial regions, but this conclusion has been repeatedly presented by Fernández and Marques (2017) and Menegotto & Rangel (2018) among the others (I'm teaching that at my marine biology courses...). Thus the authors cannot claim this as a novel finding.

Response: Thank you for your comment, and I fully agree that the idea that bimodal diversity patterns may result from unequal sampling effort has been discussed in the literature, including by Fernández & Marques (2017) and Menegotto & Rangel (2018), both of whom have already been cited and discussed in the manuscript. Please refer to the Introduction (Lines 140–141) and Discussion (Lines 471–483).

Thus, I would like to clarify that the manuscript does not claim this as a novel finding. Rather, referred to these studies explicitly as part of the established context in which the results are interpreted. The intention was to confirm and quantify this effect at a global, depth-resolved scale using a harmonised, multi-phyla and standardised dataset, rather than to propose the explanation as novel. For this, not only the bias-corrected response curves generated (Supplementary Figures 6 and 8), but Kernel estimations, Anderson–Darling (Ad-test) and Hartigan's Dip (Dip-test) tests were also applied for all depths (Supplementary Table 4), as a novel depth-explicit approach.

To avoid any misunderstanding, I have revised the relevant section of the Discussion to explicitly acknowledge these prior works and to clarify that our results are consistent with and extend their findings across the full oceanic depth gradient “Our findings are consistent with previous studies showing that reduced sampling in equatorial regions can obscure latitudinal richness patterns, and here this understanding is extended by demonstrating that such sampling biases persist across the full oceanic depth gradient. Using extensive occurrence data and rarefied response curves, our analyses further support earlier conclusions that the apparent bimodality in global richness patterns is largely driven by uneven sampling effort and limited data availability, as the bimodal pattern remained non-significant even after accounting for these biases.” (Lines 477–483).

8. A key finding of the present study is that deep-sea habitats are much more diverse than previously reported once sampling bias is accounted for. But again this is part of a consolidated knowledge.

Response: I appreciate the reviewer’s comment and agree that increasing evidence over the past decade has highlighted that deep-sea diversity is higher than previously thought. However, I would like to clarify that this study does not claim the high diversity of deep-sea habitats as a novel discovery, but rather provides a quantitative global assessment that systematically integrates multiple major taxa and explicitly accounts for sampling bias across all ocean depths. While prior studies have shown this pattern locally or for selected taxa, our work extends these findings by (i) applying a unified analytical framework across depth strata and taxa, (ii) quantifying the magnitude of bias correction effects on global diversity estimates (Number of Species, Expected Number of Species, Chao2 (based on presence-absence data), Abundance-based Coverage Estimator (ACE), and Inverse Simpson (species evenness), and (iii) identifying geographic regions and depth intervals where undersampling remains most pronounced.

The Discussion is now revised to emphasise that our contribution lies in the global synthesis and methodological standardisation, rather than in proposing the deep-sea diversity pattern as new knowledge “While many other studies have noted the potential for high deep-sea biodiversity, this study reinforces this understanding by providing a quantitative, bias-corrected, and taxonomically broad global assessment that extends previous findings across all ocean depths.”(Lines 430–433).

9. In addition, the study focuses on Animalia (and primarily on several major phyla). Other components of marine biodiversity – such as protists, fungi, flora (algae, seagrasses), and prokaryotes – are not included. This is understandable given data availability and the already enormous scope, but it means the “global marine biodiversity” assessed is not truly the entirety of biodiversity (it’s essentially the multicellular animal biodiversity). Some patterns (especially for microbial diversity, which can differ) are not captured. The authors might consider noting this as a scope limitation for clarity.

Response: You are totally correct, and thank you for pointing this out. The title, abstract, and methods were revised to specify “marine animal biodiversity.” A sentence was added in the Methods noting that microbial, algal, and fungal diversity were outside this study’s scope due to data limitations “Non-animal kingdoms, including Bacteria, Chromista, Plantae, Protozoa, Fungi, or Viruses, were outside this study’s scope. This scope limitation reflects current data availability and the already large scale of the analysis. Readers should note that some biodiversity patterns, particularly those of microbial and non-animal taxa, are not captured here.” (Lines 168–172).

10. The Human Impact Index correlating with mesopelagic diversity might only partly reflect that regions with high human presence (e.g., North Atlantic) were better sampled rather than genuinely higher in intrinsic diversity. The models did include sampling effort as a covariate in GLMs, which might help, but interpreting human impact causally is tricky.

Response: Thank you for this important point. Yes, the correlations between the Human Impact Index and mesopelagic diversity may partly reflect sampling intensity rather than intrinsic ecological patterns.

To address this, the Abstract was revised: “Shallow-water gamma species richness was associated with temperature, whereas richness patterns in deeper waters were strongly related to higher human impact due to increasing sampling efforts, and nitrate concentrations (a descriptor reflecting deep-sea organic matter remineralisation). Alpha diversity was best explained when all geocological variables were included in the model.” (Lines 33–37).

Revised the Results: “In the mesopelagic zone, human impact, which might reflect more sampling efforts, was the strongest predictor of species counts, while nitrate best explained ES50, with temperature and intercept models also competitive (Supplementary Tables 7 and 8)” (Line 367).
Revised the Discussion: “However, human influence, likely through increased sampling effort, was correlated with reported species richness in mesopelagic zones. Regions with greater human presence are typically better sampled, which may partly explain the observed association.” (Lines 531–535).

11. Similarly, temperature and productivity can covary with latitude. The authors have tried to disentangle these with their models (and by separating depth strata), but caution is warranted in attributing causation.

Response: Yes, temperature and productivity covary with latitude. Our models explicitly account for this collinearity, and the results are now interpreted as associative rather than strictly causal. This statement was revised, and a clarification has been added to the Discussion “The need for depth-specific analyses that inform targeted conservation and monitoring strategies is further highlighted by the differing environmental correlates of reported species richness across depths, such as temperature in shallow waters and nitrate concentration in deep seas. Nevertheless, these associations must be interpreted with caution because variables such as temperature and latitude covary strongly, limiting our ability to infer causation.” (Lines 569–571).

Thus, my issue here is that I do not see take home messages that are new nor producing a real improvement of the present knowledge.

Response: This assessment may overlook several substantive advances provided by the present study. While some broad patterns have been noted previously, this study provides a significant advance by integrating the largest quality-controlled global dataset assembled to date (184,141 species, compared with 11,567 in Tittensor et al. and ca. 65,000 in Chaudhary et al.). The depth-explicit patterns were analysed, and the rarefaction (ES50) method was applied to reduce sampling bias. Further significant distributional tests (Anderson–Darling for normality and the Dip test for modality) were applied on sampling effort, reported number of species, and ES50 to evaluate normality and modality of the distribution patterns per depth, calculating several other species richness estimators including Chao2, ACE, and Inverse Simpson considering both only presence absence data, abundance data, considering rare and even species “Chao2 species richness was estimated from presence–absence data using the specpool() function, providing an asymptotic estimate of total species richness that accounts for rare species based on the number of species occurring in only one or two samples. Species richness was also estimated using the ACE via the estimateR() function, which provides an asymptotic richness estimate emphasising rare species based on their individual numbers, rather than solely on presence–absence data. The Inverse Simpson index using the diversity() function with index = "invsimpson" was also estimated, accounting for both species richness and evenness while giving greater weight to abundant species, and finally species-richness drivers were modelled using both GAMs and GLMs, with ecological variables extracted separately for each depth class—an approach not implemented in prior global assessments. These methodological innovations allowed quantify spatial data gaps, identify under-sampled regions, and generate depth- and taxa-specific insights into the ecological correlates of biodiversity, thereby providing a clear and meaningful improvement over existing knowledge. Importantly, this knowledge-gap analysis provides a crucial global baseline that can directly support upcoming assessments, including the second IPBES Global Assessment and other science–policy processes.”

Other concerns:

1. The spatial and categorical structuring of the data classified available information into three depth zones – shallow (0–200 m), mesopelagic (200–500 m), and deep-sea (500–11,000 m). But mesopelagic is deep and the classification presented is thus misleading. In addition aggregating data in this way means considering 5%-8% and 87% which makes extremely unbalanced the assessment in terms of robustness of results.

Response: Agree, the mesopelagic zone is considered part of the deep ocean, if we refer to some textbooks which consider the depths below 200 m as the deep sea. However, the classification follows the World Register of Deep-Sea Species (<https://marinespecies.org/deepsea/>), which defines deep-sea habitats as starting at 500 m, where environmental variables become relatively stable. Thistle, D. The deep-sea floor: an overview. *Ecosystems of the World* 5–38 (2003), and Gage, J. D. & Tyler, P. A. *Deep-Sea Biology: A Natural History of Organisms at the Deep-Sea Floor*, (Cambridge University Press, 1991) also referred that the deep sea starts at the shelf breaks, which in some regions, such as Antarctica, is at 500m. This classification allowed ecologically meaningful comparisons across global datasets. Despite differences in area coverage among zones, all depth zones were analysed separately, and GLM and GAM models incorporated sampling effort to ensure robust and comparable results.

Additionally, Supplementary Figure 7 presents species counts per equal-size depth bands (10 m for shallow and mesopelagic zones, 100 m for the deep sea) to illustrate richness patterns at a finer resolution.

2. Seven major taxa (Annelida, Arthropoda, Chordata, Cnidaria, Echinodermata, Mollusca, Porifera) were analyzed separately in addition to the overall dataset. The choice of focusing on these dominant animal phyla is understandable yet questionable as there are 35 marine phyla (here only 7 considered).

Response: Here, seven major animal taxa (Annelida, Arthropoda, Chordata, Cnidaria, Echinodermata, Mollusca, Porifera) are represented because they represent the vast majority of marine animal biodiversity records: ~76% of occurrence records and ~93% of species counts in OBIS/GBIF (Supplementary Table 3). These taxa encompass the dominant functional and ecological diversity, while the remaining 28 phyla collectively account for only a small fraction of records. This is now explicitly noted in the Methods: “For taxon-specific analyses, seven dominant taxa, including Annelida, Arthropoda, Chordata, Cnidaria, Echinodermata, Mollusca, and Porifera, were selected, representing ca.76% of occurrence records and ca. 93% of species in OBIS and GBIF, while the remaining phyla were excluded due to their limited data availability.” (Lines 191–195).

3. The use of ~800,000 km² hexagonal cells and 5° latitude bands is pragmatic for a global view, but it is a coarse spatial grain. Finer-scale patterns (e.g., coastal reef hotspots or seamount clusters) might be obscured when averaged over such large areas. For instance, the biodiverse Coral Triangle region might not register as a hotspot in this analysis because it is under-sampled and because any given hexagon there may include more open ocean than reef. Thus, some nuance is lost at local scales. The authors could perhaps discuss this scale issue, though it does not invalidate the broad findings.

Response: Thank you for your thoughtful observation. I acknowledge that using ~800,000 km² hexagonal cells and 5-degree latitudinal bands is a coarse spatial resolution, which may obscure fine-scale patterns such as coastal reef hotspots or seamount clusters. Smaller hexagons would increase the number of cells with zero records, complicating visualisation and reducing statistical robustness, particularly for GLM and GAM analyses. This resolution was chosen to balance global coverage, sampling completeness, and model reliability. While local nuances (e.g., the Coral Triangle) may be underestimated, the approach provides a robust overview of broad-scale biodiversity patterns and knowledge gaps.

To address your comment, this scale limitation was added to the Methods: “The chosen spatial resolutions of ~800,000 km² hexagonal grids balance global coverage, sampling completeness, and model reliability (reducing the number of grids with zero occurrences), while also improving map visualisation and colour distinction. However, this coarse scale may obscure finer local patterns, such as coastal reefs or seamount hotspots, and this limitation should be acknowledged.” (Lines 217–221).

In conclusion, there are merits in the present work, especially in quantifying Global Data Gaps (we know there are gaps but we need better estimates), yet this result is not sufficient to guarantee the publication in a high-ranking journal as Nature Communications.

Response: This assessment may overlook several substantive advances provided by the present study. While some broad patterns have been noted previously, this study provides a significant advance by integrating the largest quality-controlled global dataset assembled to date (184,141 species, compared with 11,567 in Tittensor et al. and ca. 65,000 in Chaudhary et al.). The depth-explicit patterns were analysed, and the rarefaction (ES50) method was applied to reduce sampling bias. Further significant distributional tests (Anderson–Darling for normality and the Dip test for modality) were applied on sampling effort, reported number of species, and ES50 to evaluate normality and modality of the distribution patterns per depth, calculating several other species richness estimators including Chao2, ACE, and Inverse Simpson considering both only presence absence data, abundance data, considering rare and even species “Chao2 species richness was estimated from presence–absence data using the `specpool()` function, providing an asymptotic estimate of total species richness that accounts for rare species based on the number of species occurring in only one or two samples. Species richness was also estimated using the ACE via the `estimateR()` function, which provides an asymptotic richness estimate emphasising rare species based on their individual numbers, rather than solely on presence–absence data. The Inverse Simpson index using the `diversity()` function with `index = "invsimpson"` was also estimated, accounting for both species richness and evenness while giving greater weight to abundant species, and finally species-richness drivers were modelled using both GAMs and GLMs, with ecological variables extracted separately for each depth class—an approach not implemented in prior global assessments. These methodological innovations allowed quantify spatial data gaps, identify under-sampled regions, and generate depth- and taxa-specific insights into the ecological correlates of biodiversity, thereby providing a clear and meaningful improvement over existing knowledge and illustrating how biodiversity patterns evolve as more data become available

Furthermore, science-policy platforms such as IPBES will particularly benefit from this work, especially with the second global assessment currently underway, as it offers robust data and gap analyses to inform decision-making processes. The informed decisions support conservation, promote data sharing in underrepresented regions such as African countries.

Reviewer #2 (Remarks to the Author):

The Author presents a very interesting paper on sampling biases in biodiversity data of marine species across all the ocean column. After compiling and curating a database of marine species occurrences, the Author calculated the number of occurrence records, the number of unique species, the expected species richness per 50 individuals, the Chao2 metric, the abundance-based coverage estimator, and the inverse Simpson metric. The results show that there are many important gaps and biases in the distribution of marine species data. The paper is well written, the methods are correctly implemented, the results are sound, and the conclusions are based on the results. However, there are some issues that should be clarified. In consequence, I cannot recommend the publication of the manuscript in the present state.

Response: Thank you very much for the positive assessment of this study and the constructive and valuable feedback. Your suggestions on the text and on the figures improved the manuscript significantly. I have addressed all your comments one by one, clarified the methods and results,

and revised the manuscript accordingly. I hope these changes resolve your concerns and improve the clarity and quality of the manuscript.

L14: Species hotspots are not clear enough. It should refer to species distribution hotspots.

Response: Done, “species hotspots” was replaced with “species distribution hotspots” (Line 14).

L21: Please indicate how the data were modelled. The abstract lacks information about the methods. It also lacks information about what proportion of the total biodiversity the 184 thousand species represent.

Response: The abstract was revised to briefly describe the main modelling approaches. This sentence was added to the Abstract: “Generalised Linear Models (GLMs) and Generalised Additive Models (GAMs) were applied to assess how geocological predictors and human impact influence species counts and ES50, accounting for sampling effort and spatial autocorrelation” (Lines 25–26).

This statement was added to the Abstract to address your comment, “The 184,141 species included here represent approximately 87% of the existing accepted species in the World Register of Marine Species (WoRMS) and 91% of the OBIS databases” (Lines 23–25).

L124: Why were the ocean polygons modified? What modifications were applied?

Response: The missing information has been clarified and added in the Methods: “Base polygons of the global oceans and seas were initially extracted from the Marine Regions (www.marineregions.org)³⁵. The extracted ocean polygons did not cover the whole ocean area. Thus, they were modified to ensure complete and accurate coverage of all ocean and enclosed sea boundaries for extracting species occurrence data. Modifications in QGIS 3.36.2 included adjusting boundaries, merging or splitting polygons where necessary, and correcting gaps or overlaps to fully represent the study area from -90°S to 90°N and -180°W to 180°E (Supplementary Figure 1)” (Lines 58–59).

L134: What is the accuracy of the species depth? Is it 50 m higher than that accuracy? In other words, what was 50 m selected as the threshold?

Response: The 50 m depth margin was selected as a conservative and recommended threshold to account for potential inaccuracies in reported species depths and in the bathymetry layer (<https://github.com/iobis/obistools?tab=readme-ov-file#check-depth>). The 50 m margin ensures that minor discrepancies do not result in unnecessary exclusion of valid records while flagging truly inconsistent values. The sentence was revised in the Methods to clarify it “or if the reported depth exceeded the maximum bathymetry for that location, with a depth margin threshold set at 50 m”, and the citation to the GitHub was also added in the (Lines 178–179).

L137: It should make clear across the manuscript (even in the title) that the analyses only include animals, not all marine groups.

Response: Thank you very much for this great point. It is now clarified throughout the manuscript, including in the Title, Abstract, Methods (Line 166), Results (Line 295), Discussion (Line 442), and all the Figure legends, that the analyses focus exclusively on marine animals or Kingdom Animalia.

L144: How many records do not have depth information?

Response: The global number of occurrences with no depth information was 3,359,550. This information was already given in Supplementary Table 2.

Now, this information was added to the main text in Methods: “A total number of 3,359,550 occurrence records lacking depth information were excluded from depth-specific analyses (Supplementary Table 2).” (Line 190)

L152-153: Please, rephrase. Do you mean that the mean, maximum, and minimum values were obtained for all variables?

Response: Yes, the sentence was rephrased for clarification: “For all ecological variables, mean, maximum, and minimum values were obtained for both pelagic layers (shallow water and mesopelagic) and benthic layers (deep-sea)” (Lines 202–203).

L155: I do not understand what you mean by 5-degree latitudinal bands. Why bands of 5 degrees? Please provide a justification.

Response: 5-degree latitudinal bands refer to geographic zones spanning 5 degrees of latitude (e.g., 0–5°, 5–10°, 10–15°, etc.).

In this study, 5-degree bands were used because this spatial resolution provides an effective balance between ecological relevance and statistical robustness. Finer bands (e.g., 1°) produced highly fragmented species distributions with many empty cells, reducing statistical analyses' power, while much broader bands (e.g., 10°) can obscure meaningful biogeographic gradients and environmental variation. The 5-degree interval is widely used in macroecological and biogeographic studies because it (i) captures regional-scale patterns in species richness and environmental change, (ii) allows sufficient sample size per band, and (iii) minimises spatial bias associated with uneven sampling effort across latitudes. Therefore, 5-degree bands offer an appropriate compromise for detecting latitudinal trends while retaining statistical reliability." This approach follows previous global-scale studies of marine biodiversity patterns (e.g., Chaudhary et al. 2016, Saeedi et al. 2029, and Tittensor et al. 2010).

A sentence justifying the use of 5-degree latitudinal bands was added to the Methods:

“Additionally, 5-degree latitudinal bands were used because this resolution is widely used in global marine biodiversity research 17,29,47 and offers a practical balance between ecological relevance and statistical robustness, providing adequate sample size per band while minimising spatial bias from uneven sampling across latitudes.” (Lines 221–223).

L156: It is not clear to me how the continental shelves were obtained.

Response: The extracted bathymetry raster layer was used to extract grids which had depth from 0 to 140 m as continental shelves. The sentence is revised in the manuscript in the Methods: “The continental shelves were derived from the bathymetry raster layer by selecting all grid cells with depths from 0 to ≤ 140 m.” (Lines 205–206).

L162: What is the size of the hexagon? 5-degrees? It is not clear.

Response: This information was already in the manuscript; each hexagon cell is approximately 800,000 km² (Line 215).

L66: Please, explain in detail the 'expected species richness per 50 individuals'. It should be indicated at least that ES50 is the expected number of marine species in a random sample of 50 individuals. Please also explain how the extrapolation is performed.

Response: Thank you very much for this thoughtful comment. The missing information is now added to the Methods: “The ES50 method in the ‘vegan’ R package was used to estimate species richness, where ES50 represents the expected number of species in a random sample of 50 individuals in both hexagonal cells and 5-degree latitudinal bands. The extrapolation is performed using individual-based rarefaction: a rarefaction curve is generated from the observed species individuals, and the expected number of species for a standardised sample of 50 individuals is calculated. For each species, the probability of being represented at least once in a random subsample of 50 individuals is computed, and these probabilities are summed across all species”. (Lines 229–235).

L166-168: Similarly to the previous point, please define the Chao2, the Abundance-based Coverage

Estimator, and the Inverse Simpson metrics. Not all readers will be familiar with these methods.

Response: The Methods section is now expanded to define these biodiversity estimators explicitly, providing brief explanations and appropriate references: “Chao2 species richness was estimated from presence–absence data using the `specpool()` function, providing an asymptotic estimate of total species richness that accounts for rare species based on the number of species occurring in only one or two samples. Species richness was also estimated using the ACE via the `estimateR()` function, which provides an asymptotic richness estimate emphasising rare species based on their individual numbers, rather than solely on presence–absence data. The Inverse Simpson index using the `diversity()` function with `index = "invsimpson"` was also estimated, accounting for both species richness and evenness while giving greater weight to abundant species” (Lines 238–244).

L173: Please, provide the parameters for the kernel estimator.

Response: Thank you for this suggestion. The missing information is now added to the Methods: “Kernel density estimation (KDE) was applied using a Gaussian kernel to visualise uni-, bi-, or multi-modal patterns, overlaid on histograms of 5-degree latitude bins and 10 m depth bins. The bandwidth was set to 10° for latitude and 20 m for depth to balance smoothing while preserving patterns in regions with limited sampling efforts” (Lines 247–250).

L180: Why are GLM applied to 5-degree bands and GAM to hexagonal cells? Please provide a justification.

Response: The missing information was added to the Methods: “GLMs, applied to gamma species richness (species counts per 5-degree latitudinal bands), enabled robust estimation of linear relationships while accounting for variation in sampling effort via the number of records per band. GAMs, applied to alpha species richness (species counts per hexagon), captured complex, non-linear responses and finer spatial patterns, including smooth terms for environmental gradients and spatial autocorrelation. The combination of GLM and GAM ensures the detection of simple linear effects at broad scales and more complex spatially structured effects at finer scales.” (Lines 261–267).

It would be useful for a better interpretation of the results to have a map with the limits of the depth zones. This will help to interpret the maps in Figure 2. Cnidaria should be more restricted to shallow zones.

Response: This is a great suggestion, thank you very much. Following your suggestion, a supplementary figure (Supplementary Figure 3 in the revised supplementary file) was created showing the boundaries of the defined depth zones (0–200 m, 200–500 m, and 500–11,000 m). You are right, the majority of the occurrence records are indeed concentrated in shallow zones in Cnidaria.

L241: The highest number of occurrence records and species counts in shallow waters is not reflected in Figure 1.

Response: Figure 1 for the Number of Records and Species Counts was revised by changing the colour scale bands to improve visualisation, ensuring that the dominance of shallow-water records and species counts is now clearly visible and reflected in the text in the Results.

Figure 3: I suggest indicating the y-axis as a subtitle of each column of plots. The y-axis can be renamed to frequency.

Response: Following your comment, Figure 3 was revised by adding subtitles above each column (Number of Records, Number of Species, and Expected Number of Species), and repeated y-axis labels were removed. Because “frequency” could be interpreted as the number of counted individuals per species (not occurrence records) and potentially mislead readers, the y-axis stayed as the “Number of Records” for the first column.

Supplementary Figure 2: The size of the points should be reduced. This will make the gaps in the point occurrences clearer.

Response: Thank you for spotting this important point in the visualisation. The point sizes were now reduced in the revised Supplementary Figure 2, which more clearly illustrates spatial data gaps and sampling densities across all ocean basins.

Reviewer #2 (Remarks on code availability):

The Author provided the code as a PDF, not as an R file. However, the users can copy the text from the PDF and extract the code.

Response: As you correctly mentioned, all the R codes were in the Supplementary File 1. However, I will upload all the R files as a zipped folder for your review, and finally, all R scripts will be publicly available as R files on GitHub and Zenodo upon acceptance of the manuscript, ensuring full transparency and reproducibility.

Once again, special thanks to both reviewers for their detailed and constructive feedback. Their input has significantly strengthened the quality, transparency, and clarity of the study.

RESPONSE TO REVIEWERS

I would sincerely like to thank both reviewers for considering reviewing the revised manuscript and for their positive feedback and evaluation of this work. I appreciate their time. The Reviewer #2 had no further comments. Thank you very much.

I have addressed all the minor comments from Reviewer #3, which have helped substantially polish the paper before publication and further improve the manuscript. Below is a detailed, point-by-point response to both reviewers. All line numbers refer to the revised manuscript using the track change option. My responses are in **bold**.

REVIEWERS' COMMENTS

Reviewer #2 (Remarks to the Author):

I am very pleased with how the author has incorporated all my suggestions. The manuscript has been significantly improved. I have no further adjustments to suggest.

Response: Thank you very much for your kind and encouraging feedback. I sincerely appreciate the time and effort you have invested in reviewing my manuscript, as well as your constructive suggestions throughout the revision process.

Reviewer #3 (Remarks to the Author):

This is a novel study that provides a robust suite of analytical/methodologic tools to assess the most recent available data and biodiversity patterns. The clear, well-organized analytical framework presented will help advance to research that aims to dive deeper into the causality of biodiversity, and as is presents a strong backbone in which to reference for data-gaps that may be distorting future researchers' interpretations. This MS offers a good source for future analyses to use & cite – as it incorporates the bias and updated records of occurrences.

Response: Thank you very much for your positive and encouraging remarks. I greatly appreciate your recognition of the study's novelty and analytical framework, and I am pleased that you find it a useful contribution for future marine biodiversity research and data knowledge gaps.

Minor comments:

- Variables in the models not well explained in the text introduction

Response: Addressing this comment and a later comment, I have expanded the Introduction to explain the variables used for modelling: lines 108 to 127 "To model species richness responses to those drivers, a comprehensive set of ecological and bathymetric variables known to influence the physiology, biology, and distribution of marine species 35 was compiled from the Bio-ORACLE database. These variables, including physical (e.g., temperature, current velocity, bathymetry), chemical (e.g., oxygen, nitrate), and productivity-related parameters (e.g., primary productivity,

radiation), were used across pelagic and benthic layers to capture bathymetrical environmental variability.

Multiple environmental drivers, such as temperature, dissolved oxygen, and primary productivity, are key determinants of species richness in shallow marine ecosystems 17,36,37. In contrast, the deep sea is characterised by extreme conditions such as high pressure, low temperatures, and limited light availability, which impose strong constraints on deep-sea life 15,38,39. Additionally, spatial features such as continental shelves and margins were derived from the Marine Regions datasets 40. Spatial features are essential for marine biodiversity because they provide environmental conditions, habitats, and connectivity, resulting in shaping species distribution and richness. Anthropogenic pressure was also incorporated using the Global Human Influence Index (obtained from NASA as a 1-km resolution global dataset integrating population pressure, land use, infrastructure, and accessibility to quantify human impact) 41, enabling an integrated assessment of environmental and human drivers of biodiversity patterns. This approach reveals previously unrecognised marine biodiversity patterns and knowledge gaps, bringing critical insights for biodiversity monitoring, conservation planning, and sustainable ocean management in the face of accelerating environmental change.”

- Figure 5 Fix typo in Temperature reference. In the plot “TheMean” and “Themperture” in correlation plots. Additionally, the geo-ecological data variables should be written out either on the plots or in the captions.

Response: Thank you so much for spotting this mistake. I have revised Figure 5 and changed “TheMean” to “TemMea” and “Temperature” to “Temperature”. I have also revised the supplementary file and corrected this typo mistake through the supplementary file.

- Figure 5- why is the nitrate relationship positive in the deep sea (i.e. increased nitrate with increase species) vs negative in the shallow and mesopelagic?

Response: In response to your comment, I have revised and added this paragraph to the Discussion, lines 331 to 337: “Nitrate in the deep sea likely serves as a proxy for water-mass characteristics and remineralisation processes, rather than a direct driver of species richness 36. Due to the slow circulation of deep waters, inorganic nutrients such as nitrate accumulate at depth as a result of organic matter decomposition, whereas they are rapidly utilised and depleted in surface waters 36,88. Accordingly, Figure 5 showed a negative relationship between nitrate and species richness in shallow and mesopelagic layers, but a positive relationship in the deep sea, highlighting a shift in the underlying ecological processes across depth gradients.”

- A more in depth definition of the Global human influence index is needed, linking to NASA Earth Data is insufficient for readers that have no familiarity with this. I also think this should be addressed early on in the MS.

Response: I have added this sentence to the Introduction describing the Human Influence Index in the Introduction: lines 121 to 125 “Anthropogenic pressure was also incorporated using the Global Human Influence Index (obtained from NASA as a 1-km resolution global dataset integrating population pressure, land use, infrastructure, and accessibility to quantify human impact) 41, enabling an integrated assessment of environmental and human drivers of biodiversity patterns. ”, and explained more in the Methods: lines 426 to 431 “The Global Human Influence Index (HII) layer was extracted from NASA Earth Data 41. The HII layer is a 1-km resolution global dataset created from nine global data layers covering human population pressure (population density), human land use and infrastructure (built-up areas, nighttime lights, land

use/land cover), and human access (coastlines, roads, railroads, navigable rivers), developed by the Wildlife Conservation Society and CIESIN..”

- Please make explicit how the sampling bias is accounted for here, referenced in Figure 6, and throughout the manuscript. Helpful to link to the ES50 analysis.

Response: I added a clarification on how the sampling bias was calculated in the Introduction: lines 101 to 104 “Sampling biases were addressed using rarefaction-based species curves and standardised species richness estimates (ES50), which control for differences in sampling effort by estimating species richness at a fixed sample size of 50 occurrences”, in the Results: lines 226-229 “When sampling biases were corrected, using standardised species richness estimates for a fixed sample size of 50 occurrences (ES50), the diversity of deep-sea ecosystems emerges as much higher than previously reported 44, challenging some earlier assumptions that deep-sea habitats are comparatively less diverse.”, and in the Figure 6 legend “for the predicted number of species and ES50”.

- Human impact should be defined in the main text, what does this consist of? Meaning?

Response: I have added this sentence to the Introduction describing the Human Influence Index in the Introduction: lines 121 to 123 “Anthropogenic pressure was also incorporated using the Global Human Influence Index (obtained from NASA as a 1-km resolution global dataset integrating population pressure, land use, infrastructure, and accessibility to quantify human impact) 41”, and explained more in the Methods: lines 426 to 430 “The Global Human Influence Index (HII) layer was extracted from NASA Earth Data ³⁷. The HII layer is a 1-km resolution global dataset created from nine global data layers covering human population pressure (population density), human land use and infrastructure (built-up areas, nighttime lights, land use/land cover), and human access (coastlines, roads, railroads, navigable rivers), developed by the Wildlife Conservation Society and CIESIN.”

- The added explanation of the metrics used (R2; i.e., Chao2, ES50) is helpful in presenting this to the broad audience of Nat Com.

Response: The descriptions of all these methods and the R packages used to calculate them were already in the Methods, please see lines 446 to 463: “These measures included: 1) the number of occurrence records, 2) the number of unique species, 3) expected species richness per 50 individuals (ES50), 4) Chao2, 5) Abundance-based Coverage Estimator (ACE), and 6) Inverse Simpson using the ‘vegan’ R package 98. The ES50 method 98 was used to estimate species richness, where ES50 represents the expected number of species in a random sample of 50 individuals in both hexagonal cells and 5° latitudinal bands. The extrapolation is performed using individual-based rarefaction: a rarefaction curve is generated from the observed species individuals, and the expected number of species for a standardised sample of 50 individuals is calculated. For each species, the probability of being represented at least once in a random subsample of 50 individuals is computed, and these probabilities are summed across all species. This approach provides a more accurate estimate of species richness while controlling for uneven sampling effort. Chao2 species richness was estimated from presence–absence data using the specpool() function, providing an asymptotic estimate of total species richness that accounts for rare species based on the number of species occurring in only one or two samples. Species richness was also estimated using the ACE via the estimateR() function, which provides an asymptotic richness estimate emphasising rare species based on their individual numbers, rather than solely on presence–absence data. The Inverse Simpson index using the diversity() function with index =

"invsimpson" was also estimated, accounting for both species richness and evenness while giving greater weight to abundant species."

- line 443-446 should be repeated in the intro as the variables assessed here.

Response: Addressing this comment and the earlier comment, I have expanded the Introduction to explain the variables used for modelling: lines 108 to 127 "To model species richness responses to those drivers, a comprehensive set of ecological and bathymetric variables known to influence the physiology, biology, and distribution of marine species 35 was compiled from the Bio-ORACLE database. These variables, including physical (e.g., temperature, current velocity, bathymetry), chemical (e.g., oxygen, nitrate), and productivity-related parameters (e.g., primary productivity, radiation), were used across pelagic and benthic layers to capture bathymetrical environmental variability.

Multiple environmental drivers, such as temperature, dissolved oxygen, and primary productivity, are key determinants of species richness in shallow marine ecosystems 17,36,37. In contrast, the deep sea is characterised by extreme conditions such as high pressure, low temperatures, and limited light availability, which impose strong constraints on deep-sea life 15,38,39. Additionally, spatial features such as continental shelves and margins were derived from the Marine Regions datasets 40. Spatial features are essential for marine biodiversity because they provide environmental conditions, habitats, and connectivity, resulting in shaping species distribution and richness. Anthropogenic pressure was also incorporated using the Global Human Influence Index (obtained from NASA as a 1-km resolution global dataset integrating population pressure, land use, infrastructure, and accessibility to quantify human impact) 41, enabling an integrated assessment of environmental and human drivers of biodiversity patterns. This approach reveals previously unrecognised marine biodiversity patterns and knowledge gaps, bringing critical insights for biodiversity monitoring, conservation planning, and sustainable ocean management in the face of accelerating environmental change."

- Line 446- 447 can you clarify why oxygen was not included in shallow and mesopelagic models?

Response: Thank you for bringing this point to my attention. I have added a supplementary Figure (Supplementary Figure 11) showing the correlation for variables in shallow water, mesopelagic, and deep sea, showing that the O₂ had autocorrelation with mean temperature in the shallow water and mesopelagic, but not in the deep sea. To clarify this, I have added this paragraph in the methods in lines 486 to 495: "Before modelling, collinearity among predictors was assessed using pairwise correlations (Supplementary GLM/GAM results). Dissolved oxygen (O₂) showed a strong correlation with mean temperature in both shallow-water and mesopelagic habitats (a correlation coefficient of 1), indicating potential multicollinearity (Supplementary Figure 11). To reduce this issue and improve model interpretability, O₂ was excluded from the analyses for these habitats. In contrast, this strong correlation was not observed in the deep-sea environment; therefore, O₂ was retained as a predictor in the deep-sea analyses. The geocological variables included in the GLM and GAM were the total area of continental shelves (for shallow waters), continental margins (for mesopelagic and deep-sea regions), ocean area, average human impact, maximum depth, mean temperature, current velocity, ice cover (for shallow waters), nitrate, primary productivity, and O₂ (for deep-sea regions). "

Minor question (more of a curiosity): This storyline details how we have more data than ever before but it's still not enough to draw definitive conclusions. I'm curious what addition of occurrence data in the last decade has been from the tropics/ and/or the global south, are we

getting better with our efforts at all? Or are the majority of increases in data collection still from the same data-rich locations? Given the extreme bias in marine biodiversity data, with equatorial areas most understudied, would it be prudent to show trends of where/when these more recent contributions occurred (perhaps in the supplement), linking to policy and references to solving the problem?

Response: Thank you so much for this interesting question, which made me do more gap analyses, generating the supplementary Figure 10, showing the low percentage of data sharing from the central tropical ocean zones (-5° to 5°), even though hundreds of thousands of occurrence records is reported from this area in the last decade, but this includes less than 2.5% of the shared data globally. I have extended the Discussion to further debate this point, and the fact that the data sharing in the last decade is mostly related to developed countries. However, we should consider the fact that we are getting better and better with data sharing even from the equatorial regions, but still, researchers and decision makers should set a high priority for promoting data sharing in those underrepresented areas. Please see Lines 250 to 262 in the Discussion: “Central tropical regions (-5° to 5°) remain persistently underrepresented despite ongoing improvements in data sharing (Supplementary Figure 10). Although hundreds of thousands of occurrence records have been contributed from these regions over the past decade, they still account for less than 2.5% of global marine occurrence data in major repositories such as OBIS. The poor tropical sampling is reported not only in marine realms but also in terrestrial habitats 4,34,47, indicating a broader influence of sampling bias on global biodiversity assessments. While data availability continues to improve, increased efforts are needed to prioritise data mobilisation in these regions.

In contrast, developed regions, particularly the North Atlantic, around Europe and North America, remain disproportionately well sampled. Most marine biodiversity data continue to originate from those regions, resulting in spatial biases that shape observed global patterns, including perceived gradients in species richness, such as low species richness in the deep sea 44, and debated hypotheses such as bimodal latitudinal species richness gradient 48,49. ”

Supplementary Figure 10. A: Total number of marine animal occurrence records mobilised to OBIS from central tropical latitudes (-5° to 5°). B: Percentage of marine animal occurrence records from central tropical latitudes (-5° to 5°) relative to the total number of occurrence records shared in OBIS per year.